# Transcriptional immune suppression and up-regulation of double-stranded DNA damage and repair repertoires in ecDNA-containing tumors

**Miin S Lin[1], Se-Young Jo[2], Jens Luebeck[3], Howard Y Chang[4,5,6], Sihan Wu[7], Paul S Mischel[8,9]\*, Vineet Bafna[3,10]\***

[1]Bioinformatics and Systems Biology Graduate Program, University of California, San Diego, La Jolla, United States; [2]Department of Biomedical Systems Informatics and Brain Korea 21 PLUS Project for Medical Science, Yonsei University College of Medicine, Seoul, Republic of Korea; [3]Department of Computer Science and Engineering, University of California, San Diego, La Jolla, United States; [4]Center for Personal Dynamic Regulomes, Stanford University, Stanford, United States; [5]Department of Genetics, Stanford University, Stanford, United States; [6]Howard Hughes Medical Institute, Stanford University, Stanford, United States; [7]Children's Medical Center Research Institute, University of Texas Southwestern Medical Center, Dallas, United States; [8]Sarafan Chemistry, Engineering, and Medicine for Human Health (Sarafan ChEM-H), Stanford University, Stanford, United States; [9]Department of Pathology, Stanford University School of Medicine, Stanford, United States; [10]Halıcıoğlu Data Science Institute, University of California, San Diego, La Jolla, United States

**\*For correspondence:**
pmischel@stanford.edu (PSM);
vbafna@ucsd.edu (VB)

**Abstract** Extrachromosomal DNA is a common cause of oncogene amplification in cancer. The non-chromosomal inheritance of ecDNA enables tumors to rapidly evolve, contributing to treatment resistance and poor outcome for patients. The transcriptional context in which ecDNAs arise and progress, including chromosomally-driven transcription, is incompletely understood. We examined gene expression patterns of 870 tumors of varied histological types, to identify transcriptional correlates of ecDNA. Here, we show that ecDNA-containing tumors impact four major biological processes. Specifically, ecDNA-containing tumors up-regulate DNA damage and repair, cell cycle control, and mitotic processes, but down-regulate global immune regulation pathways. Taken together, these results suggest profound alterations in gene regulation in ecDNA-containing tumors, shedding light on molecular processes that give rise to their development and progression.

## eLife assessment

This study of extrachromosomal DNA (ecDNA) identifies genes that distinguish ecDNA+ and ecDNA- tumors. The findings in the manuscript are **important** and the genomic analyses **convincing**. However, some of the data remain observational and the inferences would therefore be more robust with experimental validation. This manuscript could well be of relevance to biologists interested in cancer biology and gene regulation.

## Introduction

Extrachromosomal DNA (ecDNA) are large, functional, circular double-stranded DNA molecules that are enriched for oncogenes, highly amplified, and frequently observed in a wide variety of cancer types (*Turner et al., 2017*; *Kim et al., 2020*). ecDNAs lack centromeres and are asymmetrically segregated into daughter cells during cell division, driving intratumoral genetic heterogeneity, accelerated evolution, and rapid treatment resistance (*Nathanson et al., 2014*; *Lange et al., 2022*). Further, recent studies demonstrate strong positive selection for ecDNA during tumor progression (*Luebeck et al., 2023*). ecDNAs also exhibit highly accessible chromatin and altered cis- and trans- regulation, including cooperative intramolecular interactions (*Hung et al., 2021*), promoting elevated expression of oncogenic transcriptional programs (*Wu et al., 2019*; *Morton et al., 2019*; *van Leen et al., 2022*), further contributing to poor outcome for patients (*Kim et al., 2020*).

The recent development of computational tools that enable the detection of ecDNA from whole genome sequencing data, has facilitated analyses of well-curated, publicly available datasets, including The Cancer Genome Atlas (TCGA), thereby providing an important opportunity to identify transcriptional repertoires that are preferentially detected in bona fide, clinical ecDNA-containing tumors. To shed new light on the gene expression patterns that may enhance ecDNA development and progression, we examined a global transcriptional analysis of ecDNA-containing tumors.

## Results

A recent analysis utilized the tools AmpliconArchitect and AmpliconClassifier on 1921 tumors from TCGA to suggest that ecDNA prevalence ranges from 0–59.6% across multiple tumor tissue subtypes (*Kim et al., 2020*). Using AmpliconClassifier (AC), the analysis classified tumor samples into five subtypes: ecDNA(+), Breakage Fusion Bridge (BFB), complex non-cyclic, linear, and no-amplification. However, due to limitations imposed by short-read sequencing, AC may classify some ecDNA(+) structures as complex non-cyclic when breakpoints are missed. Second, BFB cycles can give rise to ecDNA formation, making discernment of the two modes of amplification difficult. To limit false-negative ecDNA classifications in the ecDNA(-) set, we treated samples with only a linear or no-amplification status as ecDNA(-), removing complex non-cyclic and BFB(+) samples from the analysis. In order to understand the transcriptional programs active in maintaining ecDNA, we selected 870 samples from 14 tumor types with at least three ecDNA(+) samples each, and compared the gene expression data of the resulting 234 ecDNA(+) and 636 ecDNA(-) samples (*Supplementary file 1B*).

### Machine learning identifies candidate genes for ecDNA maintenance

In lieu of identifying genes that are highly differentially expressed between ecDNA(+) and ecDNA(-) samples but driven by a small subset of cases (e.g. gene A in *Figure 1—figure supplement 1*), we sought to identify genes (e.g. gene B) whose expression level was predictive of ecDNA presence. We assumed that genes that were persistently over-expressed or under-expressed in ecDNA(+) samples relative to ecDNA(-) samples were more likely to be involved in ecDNA biogenesis or maintenance, or in mediating the cellular response to the presence of ecDNA.

To identify a minimal set of genes whose expression values were consistently predictive of ecDNA presence, we used Boruta, (*Kursa and Rudnicki, 2010*) an automated feature selection algorithm (*Figure 1A* and Methods). Given the unequal representation of ecDNA(+) and ecDNA(-) samples within each of the 14 tumor types, we performed Boruta on 200 datasets, each consisting of a random selection of 80% of the 870 samples (*Figure 1A*), and chose the criterion of a gene being labeled as a Boruta gene in at least 10 of the 200 trials to be selected for downstream analysis. The Boruta analysis identified a set of 408 genes with persistent differential expression, hereafter denoted as the Core gene set.

### Extending the core set with co-expressed genes

We note that the Core gene set is not a comprehensive list of discriminatory genes, using a toy example. Consider gene 'B,' a member of the core gene set, and another gene, 'C,' whose expression values across all samples are nearly identical to the expression values of core gene B. The Boruta analysis would not need to assign gene C to the core set in addition to gene B, because adding both genes incurs the same predictive power as adding one. However, either, or both genes may play an

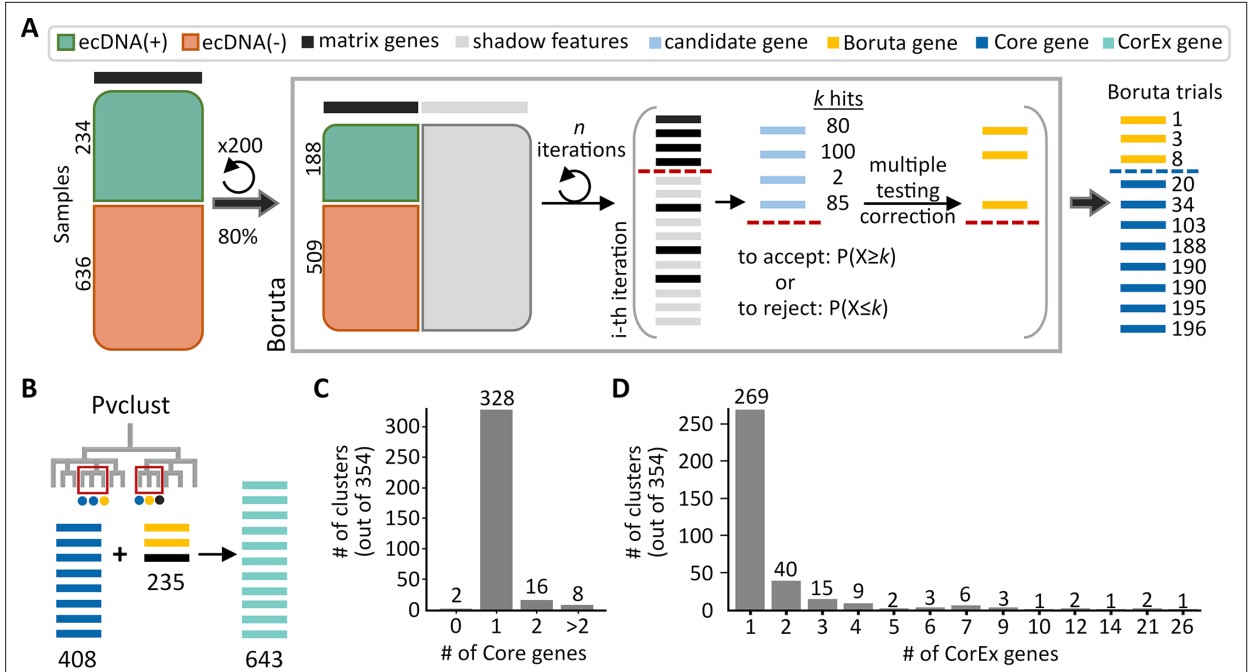

**Figure 1.** Genes predictive of extrachromosomal DNA (ecDNA) status. (**A**) The feature selection algorithm, Boruta, was applied to 200 datasets of randomly selected subsets consisting of 80% of all samples. Genes selected by Boruta in at least 10 of the 200 trials were identified as the Core set of genes (408) that were predictive of ecDNA presence. (**B**) Identification of highly co-expressed and stable gene clusters using pvclust expanded the Core set by an additional 235 genes to the final list of 643 CorEx genes. (**C**) Out of 354 clusters, the majority (344) of clusters contained one or two Core genes. (**D**) Most clusters were small, with only seven clusters containing more than 10 genes.

The online version of this article includes the following figure supplement(s) for figure 1:

**Figure supplement 1.** Cartoon illustration of the rationale for Boruta analysis.

**Figure supplement 2.** Extended co-expressed cluster characterization.

**Figure supplement 3.** Gene expression of Core versus co-expressed genes in clusters.

**Figure supplement 4.** Impact of tumor purity on CorEx gene expression.

**Figure supplement 5.** Gene Ontology (GO) biological processes enriched by up-regulated CorEx genes from eight selection criteria ranging from 5 to 200 of 200 Boruta trials.

important functional role. To correct this, we ran pvclust (**Suzuki and Shimodaira, 2006**) to cluster all gene expression values, and to identify stable clusters using multiscale bootstrap resampling (**Figure 1B**; Methods). We used an approximately unbiased (AU) confidence value of 0.95 to select the most highly co-expressed gene clusters. An AU confidence value of 0.95 represents the rejection of the null hypothesis that a group of genes fails to form a stable cluster at a significance level of 0.05. Recomputing the number of Boruta trials that members of a cluster were selected in, we selected clusters that appeared in at least 10 of the 200 Boruta trials (Methods). This resulted in the selection of 354 recurring clusters (**Supplementary file 1C**).

Notably, among the 354 clusters, only two clusters (with 14 total genes) did not contain any Core genes. As most genes do not have completely identical expression patterns, we would expect one gene to be consistently picked as a Boruta gene over another co-expressed gene. Consistent with this hypothesis, most (344/354) clusters contained only 1 or 2 Core genes (**Figure 1C**). When selecting clusters that contained at least one Core and one co-expressed gene, 53 of 71 clusters contained 1–3 Core genes (**Figure 1—figure supplement 2**), confirming that a few genes per co-expressed cluster provide sufficient predictive value, but other co-expressed genes might still play an important functional role in maintaining ecDNA presence. This is true for clusters of various sizes, including the 2-member cluster #74 and the 21-member cluster #3. In cluster #74, *CSTF1* had similar expression values to the Core gene *RAE1*, which is a mitotic checkpoint regulator implicated in tumor progression (**Kobayashi et al., 2021**; **Figure 1—figure supplement 3**; **Supplementary file 1D**). While not

necessarily increasing the predictive value, *CSTF1* is also a proto-oncogene involved in aberrant alternative splicing events (*Wang et al., 2019*). In cluster #3, 12 genes were highly co-expressed with nine Core genes (*Figure 1—figure supplement 3*; *Supplementary file 1D*), and were enriched in cell-cycle related biological processes (Methods). Importantly, the total number of genes per cluster was also small (*Figure 1D*), with only 7 of 354 clusters carrying more than 10 genes. This suggests that the Core genes have specific roles that cannot be accomplished by multiple other genes.

Summarizing, the 354 clusters contained 643 genes, which included 408 Core genes and 235 additional genes (*Figure 1B*). Together, we define these genes as the CorEx (Core + co-expressed) genes (*Supplementary file 1A*). To address the concern that the selection of CorEx genes based on bulk RNA-seq expression data could be confounded by tumor purity, we utilized a composite tumor purity score (CPE) (*Aran et al., 2015*), and observed that the ecDNA(-) samples had slightly (but significantly) lower purity than ecDNA(+) samples (*p*-value 0.0036; *Figure 1—figure supplement 4*). This is consistent with reduced detection of ecDNA in less pure samples. However, lower sensitivity of ecDNA detection would reduce the strength of the signal but not result in false positives. Indeed, when we compared the significance of CorEx gene directionality in highly pure samples (tumor purity≥0.8; n=287) versus all samples (n=870), we found a significant correlation (*Figure 1—figure supplement 4*), indicating the robustness of the CorEx set. The remaining manuscript investigates the functional properties of these genes.

## CorEx genes are better predictors of ecDNA status compared to other gene sets

We validated the relevance of CorEx genes in ecDNA presence by running cross-validation experiments (*Figure 2A*; Methods) to test the predictive power of CorEx gene expression in determining the ecDNA status of the sample. For comparisons, we used three other gene lists. The first list was a randomly chosen gene subset of identical size. For the second list, we performed a differential expression analysis using DESeq2 (*Love et al., 2014*) and picked the 643 most significantly differentially expressed genes in terms of the absolute value of their shrunken log-fold change estimate (LFC; Methods). Using the sign of the LFC value as the determinant for directionality, 240 of these genes were up-regulated, while 403 were down-regulated. Notably, only 86 of these Top-|LFC| genes overlapped with the CorEx gene set (*Supplementary file 1F*; *Figure 2—figure supplement 1*). For the third list, we used a generalized linear model (GLM) to predict 3012 genes whose expression levels were significantly associated with sample ecDNA status using a logit function after controlling for tumor subtype (Methods). Together, the three additional gene lists were denoted as random, Top-|LFC|, and GLM.

For cross-validation tests, we performed multiple random 80–20 splits of the samples to generate 200 training and test data-sets (*Figure 2A*). For each training-test data-set, a Random Forest method was used to train the predictability of the five gene lists (Methods) on the training data, and the predictive performance was tested on the test data. Expectedly, none of the gene lists was a great predictor of ecDNA status of a sample (*Figure 2B*, *Figure 2—figure supplement 2*). Nevertheless, the average of the area under the precision-recall curve (AUPRC) was higher for CorEx and Core genes (0.48 and 0.5), relative to GLM and Top-|LFC| (mean AUPRC: 0.43 each). For precision values of at least 0.7, the CorEx genes had significantly higher recall than Top-|LFC| genes (Mann-Whitney U-test p-value 4.8e-21) or GLM genes (p-value 8.5e-20). In turn, the Top-|LFC| and GLM genes were more predictive than random (mean AUPRC: 0.36). Expectedly, the predictive performance did not change when switching between Core genes and CorEx genes, because each of the non-core genes in the CorEx list had an expression pattern similar to at least one Core gene (*Figure 2B*, *Figure 2—figure supplement 2*).

To test the persistence of CorEx genes across tumor types, we re-computed Cliff's delta values *Cliff, 1993*; *Cliff, 1996* for each of the 11 TCGA tumor types that had at least 10 ecDNA(+) and at least 10 ecDNA(-) samples. The directionality of gene expression patterns was significantly similar to TCGA in each tissue type, with one exception (*Figure 2C*; *Supplementary file 1E*). The sole exception was the tumor type of Sarcoma (SARC). It is notable that the TCGA-SARC samples included many liposarcomas. In addition to containing ecDNA, liposarcoma samples are known to have extensively rearranged structures indicative of chromothripsis and neo-chromosome formation (*Garsed et al., 2014*). For other tissue types, the *p*-values against a null hypothesis of no match to the pan-cancer prediction ranged from 4.2e-12–3.5e-85 (Fisher's exact test) for the significant associations (Methods).

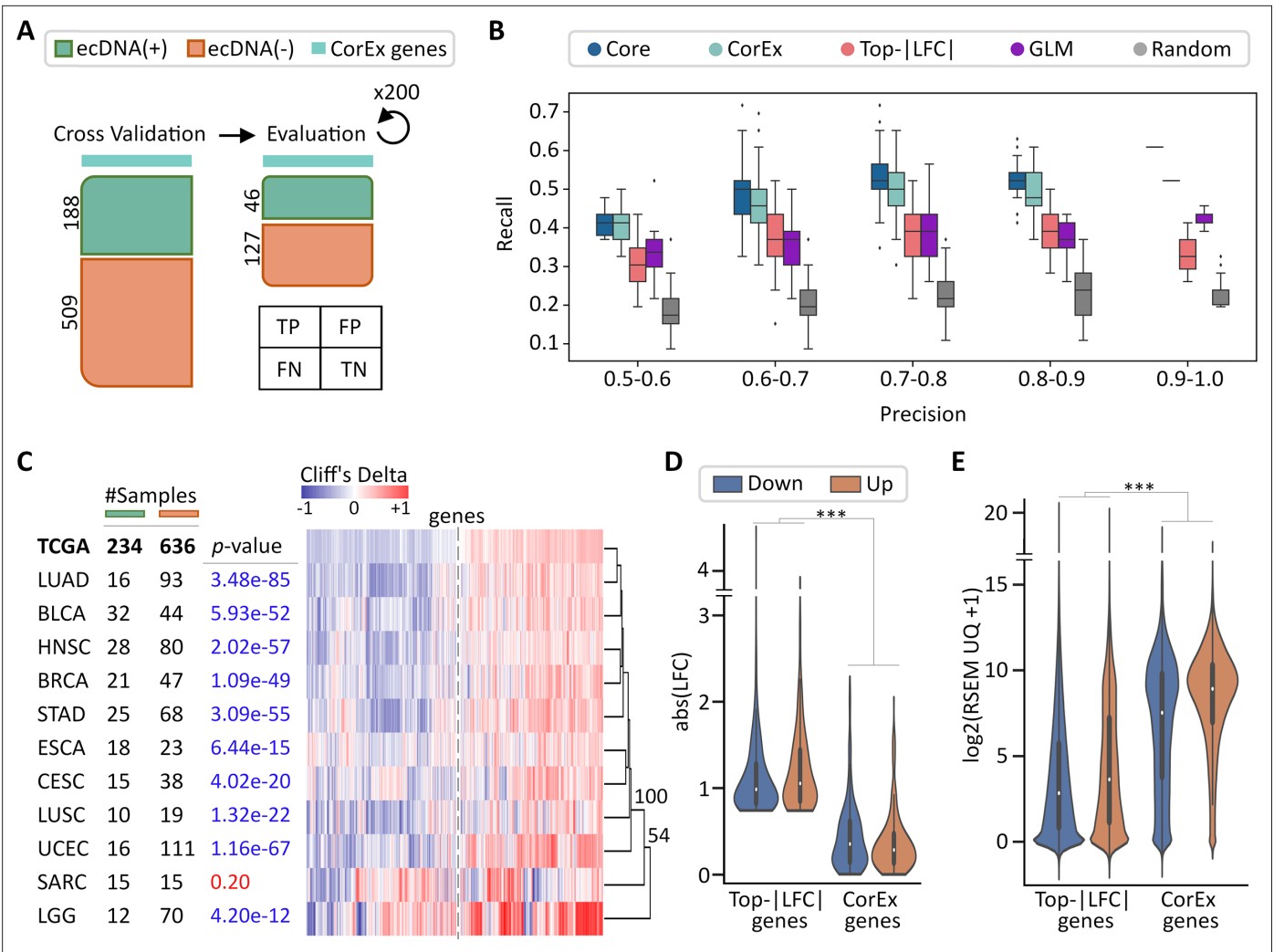

**Figure 2.** Validation of CorEx genes. (**A**) Cross-validation experiments validating the predictive value of CorEx genes. Precision denotes the fraction of predicted samples that were truly ecDNA(+). Recall refers to the fraction of ecDNA(+) samples that were predicted correctly. (**B**) For precision windows of width 0.1 and a value of at least 0.5, recall values were plotted as boxplots. The interquartile ranges for CorEx and Core genes overlap, suggesting similar predictive power. CorEx genes have higher predictive rates compared to the top 643 differentially expressed genes based on logarithmic fold changes from a DESeq2 analysis (Top-|LFC| genes), 3012 significant genes selected from a generalized linear model (GLM), and 643 randomly selected genes. (**C**) CorEx genes were consistently up- or down-regulated in ecDNA(+) samples across tumor types, with the exception of Sarcoma (SARC). Approximately unbiased (AU) p-values from multiscale bootstrap resampling are shown at the dendrogram branches. (**D**) Of the 643 Top-|LFC| genes, 240 were up-regulated while 403 were down-regulated in ecDNA(+) samples. Of the CorEx genes, 325 were up-regulated while 318 were down-regulated. The absolute log-fold change (LFC) values of the Top-|LFC| gene set was significantly greater than that of the CorEx genes (p-value 1.83e-158). (**E**) The normalized gene expression values of the CorEx genes were significantly higher than that of the Top-|LFC| gene set (p-value <2e-308). ***p-value <0.001.

The online version of this article includes the following figure supplement(s) for figure 2:

**Figure supplement 1.** Gene expression of Top-|LFC| versus CorEx genes.

**Figure supplement 2.** Extended validation of CorEx genes.

**Figure supplement 3.** Differential expression of Core genes by tumor type.

**Figure supplement 4.** Up-regulation of CorEx genes located on amplicons.

The results were similar if we tested using only Core genes (*Figure 2—figure supplement 3*). Summarizing, the 643 CorEx genes are differentially expressed across a multitude of tumor types, and have consistently higher or lower expression in ecDNA(+) samples relative to ecDNA(-) samples. These results are consistent with a pan-cancer role of CorEx genes in ecDNA biogenesis and maintenance.

The Top-|LFC| genes were also different from the CorEx genes by other metrics. Not surprisingly, the log-fold change (LFC) values of the top-|LFC| genes were higher than the LFC values of the CorEx genes (*Figure 2D*, MWU *p*-value 1.83e-158). However, much of the LFC change was due to the very low expression of the top-|LFC| genes in either ecDNA(+), or ecDNA(-) samples. In fact, the CorEx genes had higher expression in both ecDNA(+) and ecDNA(-) samples compared to the Differentially Expressed (DE) genes (*Figure 2E*, MWU *p*-value <2e-308). While the absolute log fold-change in expressions of CorEx genes between ecDNA(+) and ecDNA(-) samples was not that high (median: 0.30, mean: 0.41), it was persistent across all samples (variance: 0.14, standard deviation: 0.37).

For example, the genes *ITLN1* and *PNMT* had the second and eighth-highest absolute LFC values of 3.92 and 2.89 in the top-|LFC| list. However, their normalized expression values in most ecDNA(+) samples were low. *ITLN1* had a normalized RSEM expression value $\leq 8$ (21$^{st}$ percentile) in 210/234 ecDNA(+) samples. Similarly, the normalized RSEM expression value of *PNMT* in 223/234 ecDNA(+) samples was less than 8.5 (rank percentile: 41.1%). For *PNMT*, the differential expression was mediated by 11 ecDNA(+) samples having an expression value $\geq 11$, and 5 of the 11 samples contained *PNMT* on an ecDNA amplicon (*Figure 2—figure supplement 4*). Similarly, three samples with high RSEM contained *ITLN1* on an amplicon (*Figure 2—figure supplement 4*), partly accounting for the high |LFC| value. In contrast, the CorEx gene, *RAE1*, had a high normalized expression value in both ecDNA(+) and ecDNA(-) samples (average 9.72, rank percentile 74.3%), with a small but persistent LFC value of 0.33.

The results confirm our intuition that differential expression can arise due to multiple reasons, including low expression of the gene in a majority of samples, or the copy number amplification of a gene in a few samples. In contrast, the CorEx genes were selected based on persistent over- or under-expression in ecDNA(+) samples.

## CorEx genes primarily up-regulate three biological processes: Cell cycle, cell division, and DNA damage response

To identify enriched biological processes specific to either up-regulated or down-regulated genes in ecDNA(+) samples, we combined two metrics of effect size, Cliff's delta (*Cliff, 1993*; *Cliff, 1996*), and log fold change *Love et al., 2014* to determine the directionality of CorEx genes (*Supplementary file 1G*; Methods). The two effect size metrics were mostly in agreement in terms of directionality. Of the 7288 genes that passed the negligible effect size thresholds in both metrics, only 14 were not in concordance. This more stringent approach, in comparison to a simple directionality based on the sign of a single effect size value, was applied given that enrichment analysis on gene sets is dependent on not only the number of up- or down-regulated genes but also the degree of overlap with genes under a specific biological process term (Methods). Using this approach, among the 643 CorEx genes, 262 genes were found to be up-regulated in ecDNA(+), while 271 were found to be down-regulated (*Supplementary file 1A*; Methods). 110 genes did not make the effect size cut-off. The numbers were similar for the 408 Core genes, with 190 up-regulated, 196 down-regulated, and 22 genes not making the cut-off.

We performed enrichment analysis on gene sets to identify the Gene Ontology (GO) biological processes that are enriched in CorEx genes (Methods). Briefly, we applied a one-sided Fisher's exact test using gene sets from MSigDB (*Subramanian et al., 2005*; *Liberzon et al., 2011*; *Liberzon et al., 2015*), using a false discovery rate of 5% (Benjamini-Hochberg procedure). The UP-regulated genes enriched 187 Biological processes (*Supplementary file 1H*). Note that the GO-biological process (BP) terms are not independent, because of their hierarchical organization, and sharing of genes across different GO terms. Therefore, we used an approach similar to that used in DAVID *Huang et al., 2007* to cluster the biological processes enriched by the UP-regulated genes into 11 broad categories (*Supplementary file 1I*; *Figure 3—figure supplement 1*; Methods). The 11 categories were assigned a name using manual inspection of the constituent GO terms, or called 'Other.' The 11 categories (including 'Other') are shown in a waterfall plot to explain the contribution of each gene to a category (*Figure 3A*).

The 10 categories included expected participation of biological processes involved in (a) cell-cycle regulation (Mitotic/Meiotic Cell Cycle, G1/S, G2/M) (b) cell-division (Spindle Organization, Cell Division, Chromosome Condensation, Chromosome Segregation), (c) DNA Damage response (DNA Repair), and (d) the HOX Gene cluster. Indeed, one of the largest clusters, cluster #3, containing

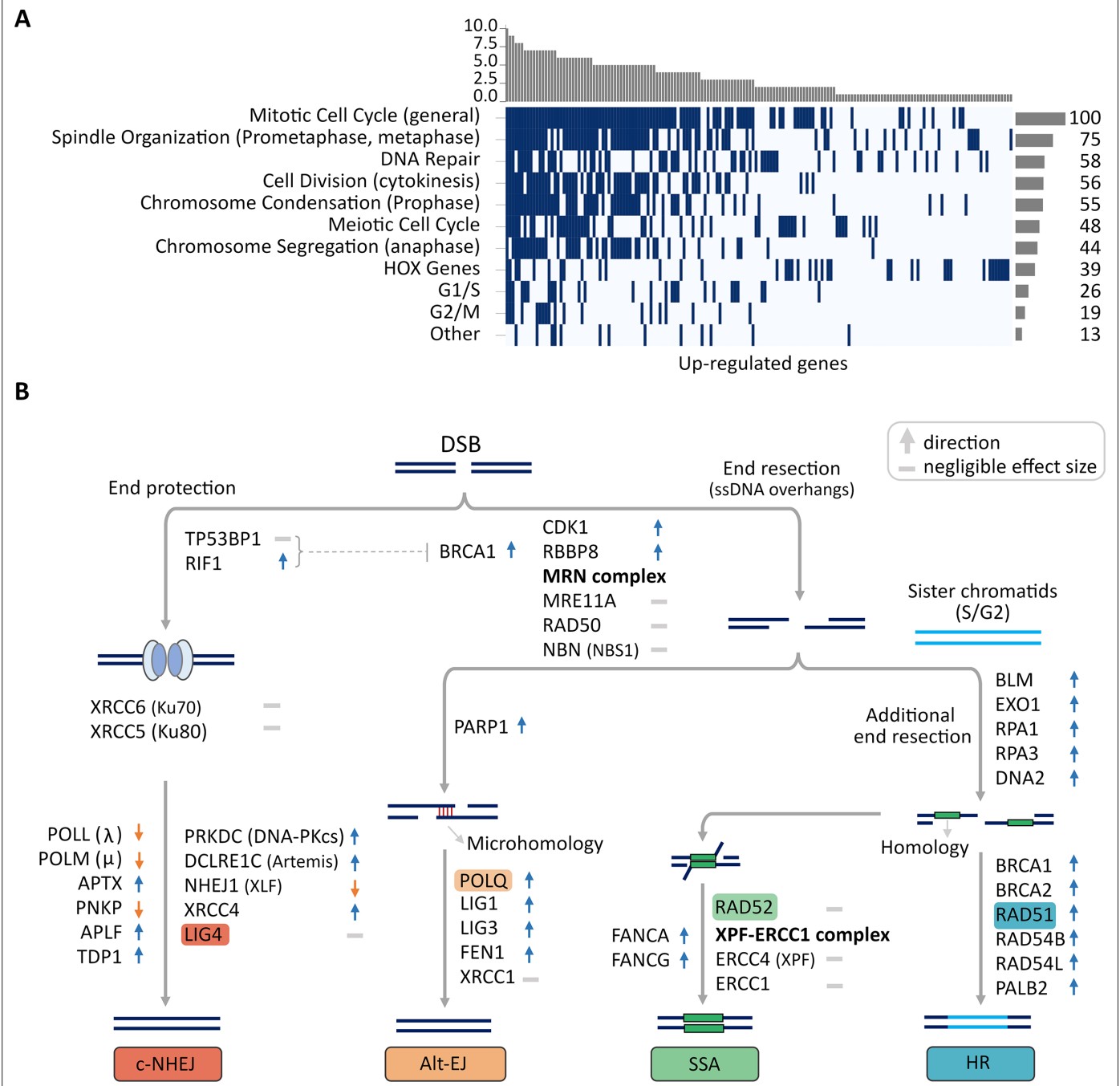

**Figure 3.** Up-regulated CorEx genes. (**A**) Gene Ontology (GO) biological processes enriched in up-regulated genes were clustered into 11 broad categories. The horizontal barplot represents the number of GO biological processes belonging to each of the 11 broad categories, while the vertical barplot represents the number of broad categories that a specific GO biological process belongs to. (**B**) Genes up- or down-regulated in processes involved in major double-strand break (DSB) damage repair pathways. Many critical genes in the classical non-homologous end-joining (c-NHEJ) pathway were down-regulated in ecDNA(+) samples relative to ecDNA(-) samples.

The online version of this article includes the following figure supplement(s) for figure 3:

**Figure supplement 1.** Biological process categories enriched in up-regulated CorEx genes.

9 Core genes and 12 co-expressed genes, was enriched in GO-BP terms related to the cell cycle (*Supplementary file 1J*). Notably, the enriched categories also included a role for the *HOX* genes with 17 members of the *HOX* family up-regulated in ecDNA(+) cancers (*Supplementary file 1I*). Many recent reports have associated *HOX* genes with cancer, including an association with their phenotypic

'hallmarks' (*Feng et al., 2021*). Genes involved in angiogenesis (*HOXA2*, *HOXC5*), genome instability (*HOXC5*, *HOXC11*), deregulating cellular energetics (*HOXA4*, *HOXC5*), and metastasis (*HOXA2*) were all up-regulated in ecDNA(+) cancers.

While the Meiotic cell-cycle was also enriched, only seven genes were allocated specifically to the group of enriched terms: *SEPP1*, *SNRPA1*, *TMEM203*, *RNF114*, *TAF4*, *TNFAIP6*, and *PTX3* (*Supplementary file 1K*). Though these genes have roles in the meiotic cell cycle, they have also been implicated in cancer and other inflammatory diseases. The spliceosomal protein SNRPA1 is a pro-metastatic splicing enhancer (*Fish et al., 2021*). Over-expression of TNFAIP6 has been associated with metastasis and poor prognosis for patients (*Zhang et al., 2021*). TMEM203 is a STING-centered signaling regulator implicated in inflammatory diseases (*Li et al., 2019*). RNF114 is a zinc-binding protein whose over-expression is an indicator of epithelial inflammation and is implicated in various tumors (*Feng et al., 2022*). TAF4, a transcription initiation factor, when overexpressed, is implicated in ovarian cancer by playing a role in dedifferentiation that promotes metastasis and chemoresistance (*Ribeiro et al., 2014*). Finally, in looking at the genes in the 'Other' category, *SHCBP1* was the only gene unique to it. A member of the neural precursor cell proliferation process, SHCBP1 is reported to promote tumor cell signaling and proliferation (*Xu et al., 2020*).

Taken together, the 11 biological process categories explain 165 of the 262 up-regulated CorEx genes, and suggest that mitotic cell-division, cell cycle regulation, and DNA damage response are the three broad categories of biological processes up-regulated to ensure ecDNA presence, along with an up-regulation of genes in the *HOX* cluster.

## CorEx genes up-regulate specific double-strand break repair pathways

The 16 enriched DNA damage response GO terms contained the terms 'double-strand break repair' and 'recombination,' but none contained the terms 'single-strand,' 'nucleotide-excision,' or 'mismatch-repair' (*Supplementary file 1I*), suggesting that the CorEx genes are largely composed of genes involved in multiple double-strand breaks (DSB) repair pathways, which include classical non-homologous end-joining (c-NHEJ), Alternative end-joining (Alt-EJ), single-strand annealing (SSA), or homology-directed repair (HDR) (*Chang et al., 2017*).

The choice of these varied DSB repair mechanisms for ecDNA presence is not well understood. We compiled and hand-curated a list of 129 genes involved in DSB repair and marked them for their role in one or more of these four pathways (*Supplementary file 1L*). Of these genes, a high number (67) were up-regulated in ecDNA(+), while a smaller number (15) were down-regulated, relative to ecDNA(-) samples. This breakdown of 129 DDR genes contrasts with an analysis using all genes where a nearly identical number of genes (5256, and 5251) were up- and down-regulated in ecDNA(+) samples, confirming that DDR genes are significantly up-regulated relative to all differentially expressed genes (*p*-value <0.0001; Fisher's exact test). When broken down to the roles of genes in individual DSB repair pathways, we found that Alt-EJ with 11 up-regulated and one down-regulated genes (*p*-value 0.0063), SSA (11 up, one down (*p*-value 0.0063)), and HR (46 up, eight down; *p*-value <0.00001) were all up-regulated. However, classical NHEJ (14 up, seven down; *p*-value: 0.19) was not significantly up-regulated in ecDNA(+) samples relative to ecDNA(-) samples (Methods, *Supplementary file 1M*, *Supplementary file 1N*).

The expression of key genes in these pathways raises the possibility of an increased role of non-classical-NHEJ processes in ecDNA development or progression, relative to c-NHEJ (*Figure 3B*; *Supplementary file 1L*). A number of genes involved in c-NHEJ were down-regulated in ecDNA-containing tumors relative to non-ecDNA tumors. These included *XLF/NHEJ1* (MWU *p*-value 2.05e-03), which is a key member of the ligase complex required for c-NHEJ; *LIG4*, another member of the ligase complex (MWU *p*-value 0.03), *PNKP*, which generates 5'-phosphate/3'-hydroxyl DNA termini required for ligation (MWU *p*-value 3.90e-06); and also, DNA polymerases $\lambda$ (*POLL*; MWU *p*-value 1.09e-21) and µ (*POLM*; MWU *p*-value 0.01), which promotes the ligation of terminally compatible overhangs requiring fill-in synthesis and promotes the ligation of incompatible 3' overhangs *Ghosh and Raghavan, 2021* in a template independent manner, respectively. This does not imply a defect in these repair processes, but rather, potentially additional or preferential utilization of alternative DSB repair pathways in ecDNA-containing tumors.

TP53BP1 is key to blocking resection and promoting the c-NHEJ pathway choice, but is displaced by BRCA1 and the MRN complex to initiate resection in the broken strands *Daley and Sung, 2014*;

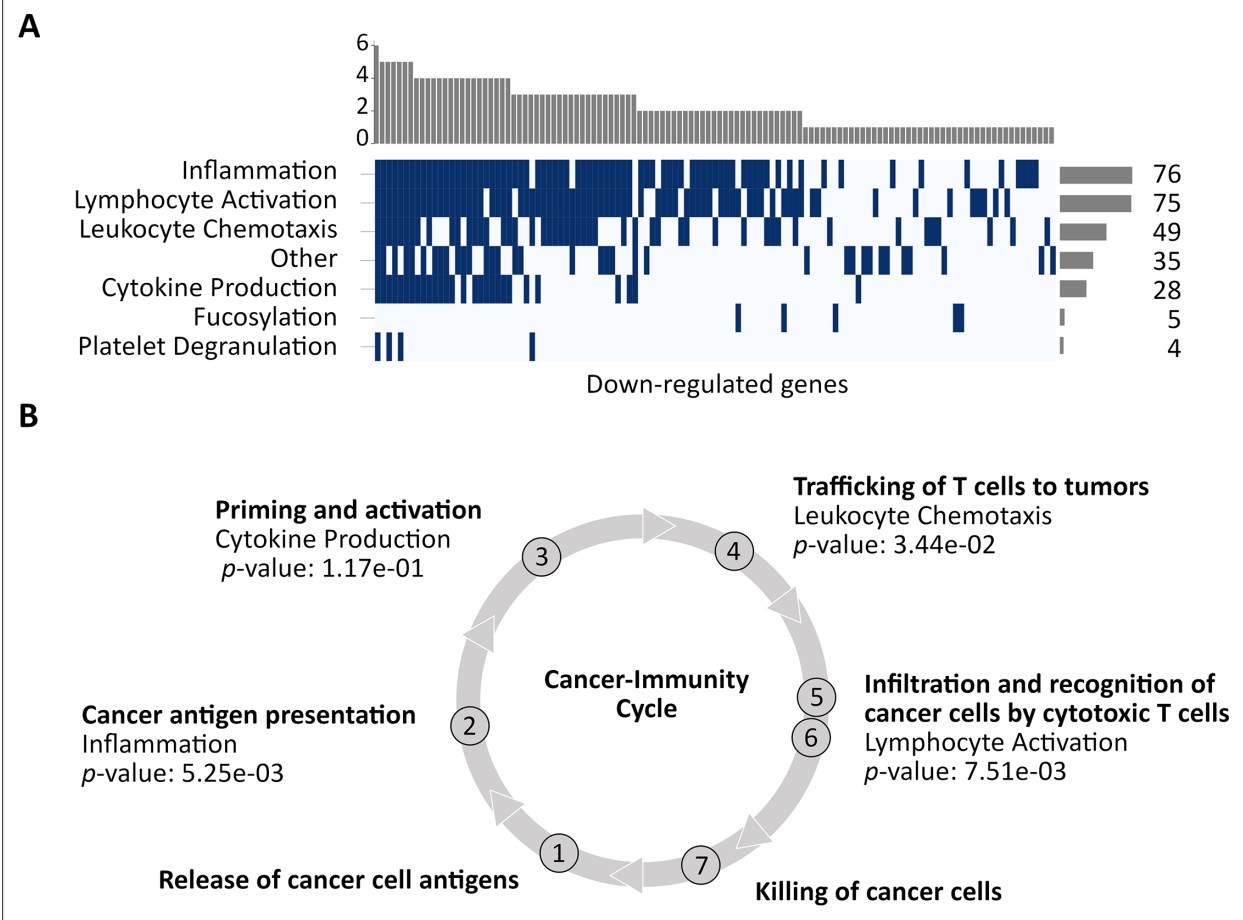

**Figure 4.** Down-regulated CorEx genes. (**A**) Gene Ontology (GO) biological processes enriched in down-regulated genes were clustered into seven broad categories. The horizontal barplot represents the number of GO biological processes belonging to each of the seven broad categories, while the vertical barplot represents the number of broad categories that a specific GO biological process belongs to. (**B**) Four of these categories map to steps in the cancer-immunity cycle. CorEx genes in three of the four categories were significantly down-regulated compared to all genes (Fisher's exact test).

The online version of this article includes the following figure supplement(s) for figure 4:

**Figure supplement 1.** Biological process categories enriched in down-regulated CorEx genes.

**Figure supplement 2.** Tumor microenvironment (TME) subtypes in extrachromosomal DNA (ecDNA)-containing tumors.

*Fouquin et al., 2017.* BRCA1 was significantly up-regulated in ecDNA(+) samples (*Figure 3B*), while *TP53BP-1* was significantly down-regulated (MWU *p*-value 0.016), although with negligible effect size (*Supplementary file 1L*). Supporting the role of alternative pathway choice for DDR, key genes in the Alt-EJ pathway, including *PARP-1*, DNA polymerase θ (*POLQ*), *LIG1*, *LIG3*, and *FEN1* were all significantly up-regulated in ecDNA(+) samples. Homology directed repair is the preferred pathway when a sister chromatid is available to act as a template. HDR is initiated by additional and extensive resection. The genes *BLM*, *EXO1*, *RPA1*, and *RPA3* which promote additional resection, as well as *BRCA1*, *BRCA2*, *RAD51*, and others that support HDR were all found to be significantly up-regulated. We can conclude that the specific pathway choice for DSB repair in ecDNA(+) may have enhanced dependence on alt-EJ and homology-directed repair pathways.

## CorEx genes primarily down-regulate immune system processes

Using a methodology similar to the analysis of the up-regulated genes, the down-regulated genes enriched 73 GO terms (*Supplementary file 1O*), and could be clustered into seven broad categories, including 'Other' (*Figure 4A*; *Supplementary file 1P*; *Figure 4—figure supplement 1*). Surprisingly, all categories were immunomodulatory. The most enriched broad category contained 75 CorEx genes relating to the Lymphocyte activation pathway. It included genes enriching 'T-cell activation'

(28 CorEx genes; *p*-value 2.58e-05), and 'Positive regulation of cell-cell adhesion' (16 CorEx genes; *p*-value 4.49e-03). Other down-regulated pathways included Cytokine activation, especially for genes in the IL-12 pathway (six CorEx genes, *p*-value 7.17e-03), TNF super-family (12 CorEx genes, *p*-value 7.24e-03), and Inflammation, including, for example, down-regulation of Toll-like receptor 2 signaling (four CorEx genes, *p*-value 4.49e-03). Finally, the broad category of Leukocyte chemotaxis was also enriched among the down-regulated genes. The chemotaxis genes include many chemokines and their receptors involved in the trafficking of T cells to the site of the tumor. The remaining down-regulated genes included four fucosyltransferases, and the category marked 'Other.' *FUT2* silencing is associated with reduced adhesion and increased metastatic potential (*He et al., 2023*). Notably, the category marked 'Other' was dominated by genes in NF-κB pathway regulation (14 CorEx genes, *p*-value 2.28e-02).

NF-κB signaling represents a prototypical, proinflammatory pathway (*Lawrence, 2009*) with multiple roles, including apoptosis. Specifically, 8 of the 14 down-regulated genes involved caspase activation (*Supplementary file 1Q*), representing the pro-apoptotic arm of NF-κB signaling. A parallel pathway for sensing endogenous ligands secreted in cell death and cancer is mediated by Toll-like receptor (TLR) proteins (*Urban-Wojciuk et al., 2019*). Remarkably, all ten TLRs were significantly down-regulated in ecDNA(+) tumors. They included TLRs expressed on the cell membrane that bind lipids and proteins as well as TLRs expressed on endosomal membranes that bind DNA. The CorEx down-regulated genes also included many involved in TLR signaling, such as *TLR3*, *CYBA*, *LYN*, and *TIRAP*.

Four of the seven broad categories mapped to facets of the cancer immune cycle (*Chen and Mellman, 2013*; *Figure 4B*). We tested if CorEx genes in these categories were more likely to be down-regulated rather than up-regulated, when compared to the non-CorEx differentially expressed genes. The Inflammation category, which mapped to the 'Cancer antigen presentation' facet, showed 18 up-regulated and 76 down-regulated CorEx genes (*p*-value 0.005, Fisher exact test, *Supplementary file 1P*). Similarly, the CorEx genes related to the 'Trafficking of T cells' facet ('Leukocyte migration and chemotaxis' category, 13 up, 49 down-regulated; *p*-value 0.03) and 'Infiltration and recognition of tumor cells by cytotoxic T cells' facet ('Lymphocyte activation' category, 23 up, 75 down-regulated; *p*-value 0.0075) were also significantly down-regulated. However, down-regulation in the 'Priming and activation' facet ('Cytokine production' category, 6 up, 28 down-regulated) was not significant at the 5% level.

As the RNA data were bulk-sequenced, transcripts were sampled from tumor cells and cells from the tumor microenvironment. *Thorsson et al., 2018* mined immune cell expression signatures to identify six immune subtypes: wound healing (C1), IFN-γ dominant (C2), inflammatory (C3), lymphocyte depleted (C4), immunologically quiet (C5), and TGF-β dominant (C6). A recent study analyzing the tumor microenvironment (TME) of ecDNA(+) vs. ecDNA(-) samples in seven tumor subtypes revealed an association of ecDNA presence with immune evasion (*Wu et al., 2022*). Our results (*Figure 4—figure supplement 2*), which used an updated version of the classification method for these ecDNA(+) samples, were broadly consistent with those from the *Wu et al., 2022*. study. Our results suggested an increase in C1 and C2 subtypes and a depletion of C3 and C6 between ecDNA(+) and ecDNA(-) categories (*p*-value 3.96e-03, Chi-squared test). Notably, the C3 (inflammatory) subtype is associated with lower levels of somatic copy number alterations, and C6 with high lymphocyte infiltration, while C1 is associated with elevated levels of angiogenic genes. These are consistent with our findings of increased somatic copy numbers, increased expression of angiogenic genes on ecDNA(+) samples, and reduced lymphocyte infiltration.

## ecDNA(+) samples carry a higher mutational burden relative to ecDNA(-) samples

In order to understand if the change in the transcriptional program was driven by mutations to the genes, we checked if ecDNA(+) samples have differential levels of mutation relative to ecDNA(-). Intriguingly, we found that the total mutation burden was significantly higher in ecDNA(+) samples relative to ecDNA(-) samples (*Figure 5A*). The result was significant also when mutations were limited to deleterious substitutions as measured by SIFT or PolyPhen2, and high-impact insertions and deletions (*Figure 5—figure supplement 1*). However, when controlling for cancer type, only glioblastoma (GBM; lower mutations in ecDNA(+)), low-grade gliomas (LGG; higher mutations in ecDNA(+)), and

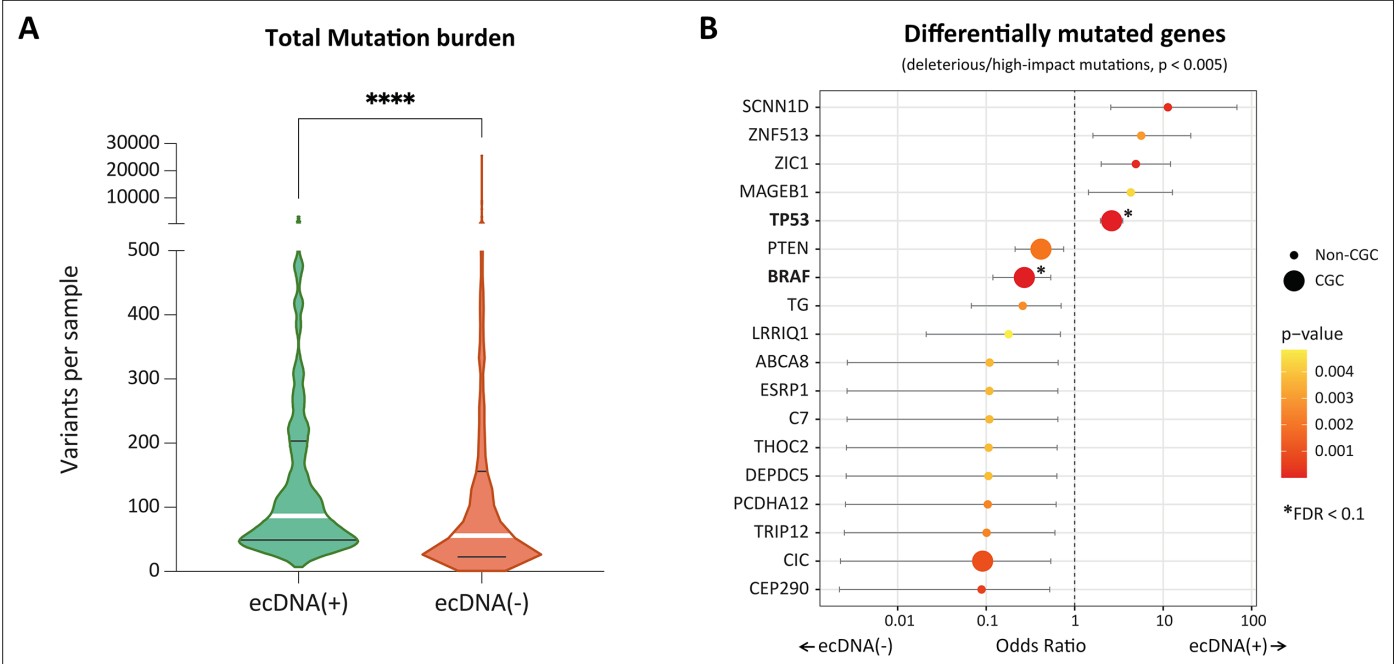

**Figure 5.** Mutational characteristics of extrachromosomal DNA (ecDNA)-containing tumors. (**A**) Total mutation burden of ecDNA(+) and ecDNA(-) samples. ecDNA(+) samples have significantly higher mutation burden than the ecDNA(-) samples (*p*-value <0.0001, Mann Whitney test). (**B**) Odds ratios of differentially mutated genes in ecDNA(+) and ecDNA(-) (*p*-value <0.005). The size of the dot indicates whether the corresponding gene belongs to the Cancer Gene Census (CGC) or not (Non-CGC). Only *TP53* and *BRAF* showed significance at the level of FDR <0.1 (Benjamini-Hochberg).

The online version of this article includes the following figure supplement(s) for figure 5:

**Figure supplement 1.** Mutation burden based on damaging mutations.

**Figure supplement 2.** Mutation burden by tumor type.

**Figure supplement 3.** Performance of XGBoost model.

**Figure supplement 4.** Unsupervised principal component analysis on gene mutations.

**Figure supplement 5.** Single base substitution (SBS) signatures in The Cancer Genome Atlas (TCGA) samples by extrachromosomal DNA (ecDNA) status.

uterine corpus endometrial carcinoma (UCEC; lower mutations in ecDNA(+)) continued to show differential total mutational burden (*Figure 5—figure supplement 2*). Of note, we did not examine the impact of a hypermutator phenotype, which could lead to high tumor mutation burden in tumors with mismatch repair deficiencies. Next, we tested if specific genes were differentially mutated between the two classes (*Figure 5B*). For deleterious/high-impact mutations, *TP53* was the only gene whose mutational patterns were significantly higher in ecDNA(+) compared to ecDNA(-) (OR 2.67, Bonferroni adjusted *p*-value 4.22e-07). BRAF mutations, however, were more common in ecDNA(-) samples and were significant to an adjusted p-value <0.1 (OR 0.27). The excess of *TP53* mutations in ecDNA(+) samples provides additional support to the hypothesis that mutations in DNA damage response or cell cycle checkpoints are important for ecDNA presence. Other genes that are differentially mutated with nominal significance (unadjusted p-value <0.005) are shown in *Supplementary file 1R*.

We also tested if a collection of gene mutations could predict ecDNA status using XGBoost (*Chen and Guestrin, 2016*), which uses an adaptive boosting of 'weak classifiers' to predict class. Here, each mutated gene was treated as a weak classifier of ecDNA status. However, the two classes could not be separated with high accuracy (*Figure 5—figure supplement 3*). An unsupervised principal component analysis did not separate the two classes either. Only the first principal component explained a significant proportion (14%) of the total variance (*Figure 5—figure supplement 4*) and did not separate the bulk of the samples. Finally, we recapitulated earlier findings that ecDNA(+) samples enrich for APOBEC activity through the presence of the mutation signatures SBS2 and SBS13 (*Bergstrom et al., 2022*; *Hadi et al., 2020*; *Figure 5—figure supplement 5*). The enrichment in *TP53* mutations was also consistent with previous findings (*Luebeck et al., 2023*). On balance, however, collections

of gene mutations did not distinguish ecDNA(+) samples from ecDNA(-) samples, at least at a pan-cancer level, in contrast to the gene expression data.

## Persistently occurring genes in ecDNA(+) samples may represent potential vulnerabilities

Any CorEx gene is either a Core gene that was selected as a feature in at least 5% of 200 Boruta trials, or be highly co-expressed with a Core gene. Because the selection criterion of 5% is arbitrary, we also tested robustness with eight other cut-offs ranging from 5-of-200–200-of-200 Boruta trials. The number of CorEx genes expectedly decreases with more stringent cut-offs. However, of the 187 GO terms that were enriched by 262 CorEx UP-genes using 10 of 200 Boruta trials as the selection criteria, 93 terms (49.7%) were enriched for each cut-off (*Figure 1—figure supplement 5*), and 155 terms (82.9%) were enriched in at least 5 of the 8 cut-off criteria. Given that our subsequent analyses utilized the hierarchy of GO terms and identified four GO-categories enriched by UP-regulated genes, the conclusions would hold regardless of the specific cut-off.

To rank CorEx genes by importance, we computed harmonic mean rank values based on three categories: (a) the average GINI importance statistic from the trained random forest models; (b) the number of Boruta trials that a gene was selected in; and (c) the number of Boruta trials (out of 200) that a gene was selected in when counting by cluster (Methods). 65 genes that were up-regulated (47 genes) or down-regulated (18 genes) had a harmonic rank lower than 3 (*Supplementary file 1A*). The next highest-ranked gene had a harmonic rank exceeding 17. These 65 genes represent the most persistent differentially expressed CorEx genes, and appeared as Core (or clustered gene) in all 200 Boruta trials. Notably, of the 24 genes most frequently expressed on ecDNA, (*Kim et al., 2020*) only EGFR, and CDK4 were included in the list of 65 genes, suggesting that the most persistent CorEx genes do not themselves appear frequently on ecDNA.

Expectedly, the high-ranked up-regulated genes impacted cell division (16 genes), cell cycle regulation (10 genes), and DNA damage response (16 genes). Only 12 of the 47 genes were not included in the gene sets of any enriched GO term. Many of these genes were from small CorEx clusters with less than three members, but we also found six genes from the HOX gene cluster (cluster #17), and another cluster of 21 genes (cluster #3). Members of cluster #3 appeared in all 200 Boruta trials; however, there were three genes all involved in cell-division (*TPX2*, *KIF2C*, and *AURKA*), each of which appeared in at least 180 Boruta trials. High expression among these three genes is associated with poor prognosis (*De Luca et al., 2006*), and due to the highly persistent nature of their differential expression across ecDNA(+) samples, they represent a possible widespread vulnerability for ecDNA(+) samples.

Intriguingly, 14 of the 18 down-regulated genes with low harmonic rank came from a single cluster (#2; *Supplementary file 1C*), and 13 of the 18 genes did not specifically enrich any specific BP ontology. Six of the down-regulated genes appeared in 180 or more Boruta trials (*CHMP7*, *XPO7*, *INTS9*, *TACR1*, *KIAA1967*, and *PCM1*). Some of these genes (*CHMP7*, *XPO7*, *KIAA1967*) are reported to be tumor suppressor genes (*Guo et al., 2021*; *Innes et al., 2021*; *Qin et al., 2015*). However, the exact functional role of down-regulating these genes in ecDNA(+) samples remains to be elucidated.

## Discussion

ecDNA is increasingly recognized as a major cause of oncogene amplification, intratumoral genetic heterogeneity, accelerated evolution, and treatment resistance, but many of the underlying processes involved in its formation, function, and progression are not fully understood. The ability to conduct multi-omic studies of well-curated, bona fide clinical tumor samples, such as the TCGA, presents an opportunity to learn about differentially regulated gene expression programs that may be involved in ecDNA biogenesis or maintenance, and in worse outcomes for patients (*Wu et al., 2019*; *Morton et al., 2019*; *van Leen et al., 2022*). Using a relatively intuitive set of principles, we have developed a machine learning approach that identifies differentially expressed, co-regulated genes in ecDNA-containing tumors, highlighting four main biological processes: non-c-NHEJ DSB repair, cell cycle, proliferation control, and immune regulation.

The GO analysis revealed three core biological processes that were up-regulated and only the immune system processes as being down-regulated. These observations strengthen the case for

targeting proteins involved in mitotic cell-division (*Von Hoff et al., 1992*), cell-cycle regulation, and DNA damage response in ecDNA(+) cancers, but also reveal roles for the *HOX* cluster of genes. Also, in this paper, we did not extensively study the role of ncRNA in the prediction of ecDNA status. We do note that *HOTAIR*, encoded in the *HOXC* locus, is independently associated with metastasis and poor outcomes (*Gupta et al., 2010*). Further experiments are needed to provide a mechanistic basis for the role of *HOX* cluster genes in maintaining ecDNA presence, as also for the involvement of ncRNA.

The DNA damage genes are broadly up-regulated in ecDNA(+) samples, especially in double-strand break repair. Within this broad category of mechanisms, our analysis suggests that alternative DSB repair pathways such as Alt-EJ are preferred relative to classical NHEJ. This is consistent with previous observations of small microhomologies at breakpoint junctions (*Kim et al., 2020*; *Sanborn et al., 2013*), and has important implications in therapeutic selection that will need to be validated in future experimental studies. We note, however, that the microhomology analyses typically study breakpoint junctions, and might ignore double-strand breaks in non-junctional sequences which could be observed, for example at replication-transcription junctions.

The down-regulated genes were primarily immunomodulatory in nature, in addition to a few persistently down-regulated tumor suppressor genes. Lowered expression of immunomodulatory genes in ecDNA(+) samples has been previously reported (*Kim et al., 2020*; *Wu et al., 2022*), but not mechanistically explained. Remarkably, the down-regulated immunomodulatory genes encompassed most aspects of the cancer immune cycle, suggesting impaired recognition of tumor DNA and proteins as foreign in ecDNA(+) tumors. Sensing of foreign DNA, including tumor DNA, is often mediated by the cGAS/STING pathway (*Samson and Ablasser, 2022*; *Sun et al., 2013*). Intriguingly, cGAS was significantly up-regulated in ecDNA(+) samples, while STING was significantly down-regulated, suggesting a role for STING agonists in intervention. Finally, in addition to the down-regulation of genes in the toll-like receptor family, we observed a down-regulation of genes involved in regulating TLR signaling pathways that were part of the CorEx list. Understanding the mechanisms of broad down-regulation of TLRs could provide insight into vulnerabilities of ecDNA(+) tumors.

Mutation data alone does not provide as clear a picture of the genes involved in ecDNA status prediction. We did observe that the total mutation burden (TMB) was higher in ecDNA(+) samples. However, that relationship is much less clear after controlling for cancer type. High TMB has been positively correlated with sensitivity to immunotherapy (*Rizvi et al., 2015*), and better patient outcomes; however, the gene expression patterns suggest that immunomodulatory genes are down-regulated in ecDNA(+) samples, and patients with ecDNA(+) tumors have worse outcomes (*Kim et al., 2020*). Notably, other results have suggested that the correlation between TMB and response to immunotherapy is not uniform, and it can vary across different tumor subtypes (*McGrail et al., 2021*). Specifically, our data is consistent with previous results which showed that Gliomas with high TMB have worse response to immunotherapy relative to gliomas with low TMB (*McGrail et al., 2021*). In general, no collection of gene mutations was predictive of ecDNA status, although mutations in *TP53* were more likely in ecDNA(+) samples, and perhaps are an important driver for ecDNA formation (*Luebeck et al., 2023*).

These results suggest that cancer cells that contain ecDNA have profound alterations in their global transcriptional patterns. Importantly, these transcriptional differences do not arise solely from genes on the ecDNAs themselves, but rather suggest that fundamental global processes involved in DSB repair, cell cycle control, and immune regulation contribute to ecDNA formation and pathogenesis.

## Methods

**Key resources table**

| Reagent type (species) or resource | Designation | Source or reference | Identifiers | Additional information |
|---|---|---|---|---|
| Software, algorithm | Amplicon Classifier | https://github.com/jluebeck/AmpliconClassifier; *Luebeck et al., 2024* | v0.4.9 | |
| Software, algorithm | Boruta | https://github.com/scikit-learn-contrib/boruta_py; *Homola et al., 2024* | 6.21.2021 | Modified to allow for early termination based on stagnant tentative counts. |

*Continued on next page*

Continued

| Reagent type (species) or resource | Designation | Source or reference | Identifiers | Additional information |
|---|---|---|---|---|
| Software, algorithm | Cliff's delta | https://github.com/neilernst/cliffsDelta: copy archived at **Ernst, 2021** | | |
| Software, algorithm | DESeq2 | https://www.bioconductor.org/packages/release/bioc/html/DESeq2.html | v1.36.0 | |
| Software, algorithm | generalized linear model | glm() function in R stats package | v4.2.0 | |
| Software, algorithm | pvclust | https://github.com/shimo-lab/pvclust; **Suzuki et al., 2019** | v2.2–0 | **Suzuki and Shimodaira, 2006** |
| Software, algorithm | CorEx | https://github.com/miinslin/ecDNA_Gene_Expression, copy archived at **Lin, 2024** | | |

## TCGA sample ecDNA status classification

Amplicon Classifier (version 0.4.9, https://github.com/jluebeck/AmpliconClassifier) classified amplicons detected in 1,921 TCGA samples into five sub-types: ecDNA, BFB, complex non-cyclic, linear, and no-amplification. When classifying a sample with multiple amplicons, the order of preference is as follows: ecDNA, BFB, complex non-cyclic, linear, and no-amplification. Given the challenges of detecting ecDNA from short read data, and to avoid possible false-negative ecDNA classifications, samples with a BFB or complex non-cyclic status which were not called ecDNA(+), were removed from the analysis. We treated samples with the linear amplification and no-amplification classifications as ecDNA(-). Of the 1921 samples, 1535 samples classified as ecDNA(+) and ecDNA(-) had RNA-seq data, including 1406 primary solid tumor samples, 95 tumor metastasis samples, and 34 primary blood-derived cancer – peripheral blood samples. Removing metastases results in a total of 1440 samples, including 243 ecDNA(+) and 1197 ecDNA(-) samples. While the set of 1440 samples represented 24 tumor types, ten of these tumor types had insufficient numbers of ecDNA(+) samples, including four tumor types with no ecDNA(+) samples. To prevent the 561 ecDNA(-) samples representing these tumors from skewing the analysis, we removed 570 samples representing tumor types with less than three ecDNA(+) samples. This resulted in a total of 870 samples representing 14 tumor types, of which 234 were classified as ecDNA(+) and 636 were classified as ecDNA(-).

## Gene expression datasets

Gene expression data for 32 studies part of the TCGA Pan-cancer Atlas was downloaded from cBio-Portal (01.05.2021) (https://www.cbioportal.org/). The cBioPortal 'data_RNA_Seq_v2_expression_median.txt' data is sourced from the file 'EB ++AdjustPANCAN_IlluminaHiSeq_RNASeqV2.geneExp.tsv' (synapse id: syn4976363). Briefly, the matrices contain batch-corrected values of the upper-quartile (UQ) normalized RSEM estimated counts data from Broad firehose (tumor.uncv2.mRNAseq_RSEM_all.txt). Missing values due to the batch effect correction process were imputed using *K*-nearest neighbors (KNN). For each tumor type, values were imputed based on gene vectors under the assumption that genes are similarly expressed between samples of the same tumor type. For genes with less than 60% of samples with missing values, values were imputed using the logarithmic (base 2) of the gene expression value plus one, and subsequently back-transformed when writing the imputed matrices to file. The resulting gene expression matrix used for the Boruta analysis described below consisted of 870 TCGA samples and 16,309 protein-coding genes (based on 'hgnc_complete_set.txt' downloaded from HGNC on 7.24.2018). To generate a RSEM raw counts matrix for the DESeq2 analysis described below, mRNAseq_Preprocess.Level_3 data was downloaded from Broad Firehose (tumor.uncv2.mRNAseq_raw_counts.txt).

## Boruta analysis

To identify a minimal set of genes whose expression values were predictive of the sample being ecDNA(+), we used Boruta (**Kursa and Rudnicki, 2010**), an automated feature selection algorithm that utilizes multiple iterations of the random forest classifier to determine the statistical significance of selected features. The algorithm is terminated when all features are categorized as 'confirmed' or 'rejected,' or until the user-defined number of iterations is reached. In our modified version of

the BorutaPy python package (6.21.2021; https://github.com/scikit-learn-contrib/boruta_py), we set the maximum number of iterations to 400, a stagnant count maximum of 5, and a tentative count minimum of 50. This translates to the termination of Boruta if 400 iterations are reached, or if the tentative count (features that have yet to be 'confirmed' or 'rejected') falls to or below 50 and these tentative features remain tentative for five iterations.

While we use a standard implementation of Boruta, the method is briefly described here for expository purposes. In each iteration, $i$, within a single Boruta trial, the input is a gene expression matrix, $M$, of dimension $r$ x $c$, where $r$ is the number of samples and $c$ is the number of tentative or confirmed features (i.e. genes). Boruta generates $c$ shadow feature vectors by random shuffling of feature vectors in matrix $M$, generating a new matrix $M'$ of dimension $r \times 2c$. A random forest classifier (class_weight = balanced_subsample, max_depth = 7) is then used to quantify the importance of each feature in separating ecDNA(+) from ecDNA(-) samples. Specifically, there are two possible outcomes for a feature: (1) if the feature scores higher than the best-scoring shadow feature, the feature is considered a 'hit,' and (2) if the feature scores lower than the best-scoring shadow feature, the feature is considered a 'non-hit.' Features are rejected after $i$ iterations, if the number of hits is not significantly higher than expected by chance, using a Bonferroni corrected $p$-value.

In order to evaluate the ability of selected features in predicting the ecDNA status of a tumor sample, we left out 20% of ecDNA(+) and 20% of ecDNA(-) samples for the hold-out testing dataset in the evaluation procedure described below, and performed Boruta on the gene expression matrix consisting of the remaining 80% of samples. However, due to the unequal representation of ecDNA(+) and ecDNA(-) samples within each of the tumor subtypes, we opted to generate 200 training (80%) and testing (20%) datasets to decrease the bias that may be introduced during random sampling. For each of the 200 datasets, a Boruta analysis was performed on the 80% training data. Features categorized as 'confirmed' were considered as Boruta genes for that specific trial. Of the 941 Boruta genes combined across the 200 trials, 408 genes were present in at least 10 of the 200 Boruta trials, and subsequently defined as the Core set of genes in downstream analyses.

## Highly co-expressed genes

To identify genes co-expressed with the core set of Boruta genes, hierarchical clustering of the 16,309 genes was performed using the R package pvclust (*Suzuki and Shimodaira, 2006*) (ver. 2.2–0; dist. method=correlation, method = ward.D2, nboot = 1000). A total of 843 significant clusters (AU >0.95) with at least one Boruta gene were selected, consisting of 1375 genes. To obtain the final list of CorEx genes, we apply a minimal count of 10 trials for the gene or 10 trials for the cluster of genes seen in 200 Boruta trials. A cluster is determined to be seen in a Boruta trial if at least one of its members is selected in the trial. This results in 354 clusters, with a total number of 643 genes, of which 408 are Core genes.

## Evaluation of CorEx genes

To evaluate a set of genes, $G$, as predictive of ecDNA presence in tumor samples, we performed cross-validation and hyper-parameter tuning on each of the 80% training datasets, and evaluated the final model on the corresponding hold-out 20% testing dataset using the scikit-learn package. Specifically, the gene expression matrix for cross-validation and hyper-parameter tuning consisted of $m$ samples from the training dataset and $n$ genes from set $G$. RandomizedSearchCV (n_iter = 50, cv = 5, scoring = f1) was first used to narrow down a wide range of hyper-parameters for the random forest classifier (RandomForestClassifier, class_weight='balanced_subsample'), and GridSearchCV (cv = StratifiedKFold(n_splits = 5, shuffle = True), scoring = f1) was then used to test every combination of a smaller range of hyper-parameters given the best parameters from RandomizedSearchCV. The hyper-parameters tuned (initial values) include the number of trees in the forest (n_estimators: np.linspace(100, 2000, num = 10)), the maximum depth of the tree (max_depth: None, np.linspace(10, 100, num = 10)), the minimum number of samples required to split an internal node (min_samples_split: (2, 5, 10)), and the minimum number of samples required to be at a leaf node (min_samples_leaf: (1, 2, 4)). The best estimator from the GridSearchCV hyper-parameter tuning was then evaluated on the 20% testing dataset, where the gene expression matrix consisted of $m$ samples from the testing dataset and $n$ genes from set $G$. Performing this procedure on each of the 200 training/testing datasets

resulted in 200 data points for each of the three metrics computed using sklearn.metrics: precision_score, recall_score, and average_precision_score (AUPR).

We performed this procedure on the following sets of genes, $G$: the 408 Core genes, 408 randomly selected genes, the 643 CorEx genes, 643 randomly selected genes, a set of 643 most differentially expressed genes based on the absolute log-fold change estimates from a conventional DE analysis using DESeq2 (*Love et al., 2014*) as described below, and the set of 3012 significant genes from the GLM analysis described below.

## Generalized linear model (GLM) analysis

We tested each of the 16,309 genes independently in a separate logistic regression model using the glm() function in the R stats package (v4.2.0), and retained genes that were significant (*p*-value 0.01). Specifically, the model was defined as glm($y \sim g_j + t$, data = $M$, family = binomial(link = 'logit')), where *y* is the response vector where $y_i$ =1 if sample $i \in \{1, ..., 870\}$ is ecDNA(+) and $y_i$ =0 otherwise, $g_j$ is the vector of expression values for gene $j \in \{1, ..., 16309\}$ in samples $i \in \{1, ..., 870\}$, *t* is the covariate vector representing the tumor subtypes of samples $i \in \{1, ..., 870\}$, and *M* is the data matrix containing values of gene expression, tumor subtype, and ecDNA status for all samples. The equation for the binomial logistic regression described above is formulated as $log\left(\frac{p}{1-p}\right) = \beta_0 + \beta_1 X_1 + ... + \beta_k X_k$, where *p* is the probability that the dependent variable *y* is 1, *X* are the independent variables, and $\beta$ are the coefficients of the model. In this case, *k*=1 represents the independent variable gene *j* and *k*=2 represents the tumor subtype covariate *t*. Of the 16,309 genes tested independently, 3012 genes were significant at *p*-value <0.01.

## Default DE analysis

We performed a default DESeq2 (*Love et al., 2014*) (R package, ver. 1.36.0) analysis to obtain shrunken maximum *a posteriori* (MAP) log-fold change estimates for effect size (i.e. LFC). Specifically, to obtain (i) LFC effect size values per gene for integration with its Cliff's delta effect size value when determining if a gene is up- or down-regulated in ecDNA(+) samples, and (ii) a list of *n* top-ranked genes by absolute value of the LFC (with application of an adjusted *p*-value <0.05 cutoff and LFC threshold of $log_2(1.1) = 0.13$) for use in comparison against genes selected as important in the prediction of ecDNA in samples. For comparisons against Core genes, *n* is set to 408, and for comparisons against CorEx genes, *n*, is set to 643.

To obtain the LFC effect size metric between ecDNA(+) vs. ecDNA(-) samples for each gene's expression, we fed as input to DESeq2 a matrix of raw RSEM estimated counts. To take into account batch effects, we included the center and platform information of samples, downloaded from synapse id syn4976363 (EB ++GeneExpAnnotation.tsv), in the design of the DESeq object:

```
DESeq_object <- DESeqDataSetFromMatrix(countData =
AC_0_4_9_TCGA_matrix, colData = coldata, design =
~batch + condition)
```

To compute results, the lfcThreshold was set to $log_2(1.1)$ for an accurate computation of *p*-values and the contrast set to c('condition,' 'ecDNA(+),' 'ecDNA(-)') to obtain the logarithmic fold change of the form $\left(\frac{ecDNA(+)}{ecDNA(-)}\right)$. By setting the lfcThreshold, the null hypothesis tested is that $|LFC| \leq \theta$, where $\theta = (1.1)$, and the alternative hypothesis is that $|LFC| > \theta$. A $log_2(1.1)$ value is chosen as the minimal value/negligible effect size threshold as it represents a 10% fold-change, and anything below this fold-change would likely not be of biological interest (*McCarthy and Smyth, 2009*). The specific commands run are as follows:

```
DESeq_object$condition <- relevel(DESeq_object$condition, ref =
"Non_ecDNA")
DESeq_object <- DESeq(DESeq_object)
results <- results(DESeq_object, lfcThreshold = log2(1.1),
contrast = c("condition","ecDNA","Non_ecDNA"))
```

To obtain the shrunken MAP log-fold change estimates, we used the lfcShrink function provided in DESeq2, using the default apeglm method for the empirical Bayes shrinkage procedure (*Zhu et al., 2019*):

```
lfcShrink(DESeq_object, lfcThreshold = log2(1.1),
coef="condition_ecDNA_vs_Non_ecDNA", type="apeglm")
```

## Up- or down-regulated genes in ecDNA(+) samples

To categorize genes as 'up-' or 'down-' regulated in ecDNA(+) samples, we integrated two effect size metrics, Cliff's delta (*d*) (*Cliff, 1993*; *Cliff, 1996*) and the DESeq2 shrunken MAPlog-fold change estimate (LFC *Love et al., 2014*). Effect size is a measure of the magnitude of deviation from the null hypothesis, and unlike *p*-values, has the advantage of not being impacted by sample size (*Romano et al., 2006*). This property is especially useful when comparing effect size values of a gene between tests where sample sizes differ. Comparing *p*-values between such tests would be invalid.

Cliff's delta, *d*, is a non-parametric measure of the separation between two distributions and ranges from –1 to 1. Given two distributions, $X = \{i_1, i_2, \ldots, i_m\}$ and $Y = \{j_1, j_2, \ldots, j_n\}$, comparisons are made between each of *m* values in *X* and *n* values in *Y*. Cliff's delta is computed as $d = \frac{\#(i>j) - \#(i<j)}{mn}$, where $\#(i > j)$ is the number of times a member of *X* is greater than a member of *Y*, and $\#(i < j)$ is the number of times a member of *X* is less than a member of *Y* (*Cliff, 1993*). A negative *d* indicates that values in *Y* tend to be higher than *X*, while a positive *d* indicates that values in *X* tend to be higher than *Y*. The magnitude of the effect size of Cliff's delta can be separated into four levels: $|d| < 0.147$ for negligible effects, $0.147 \leq |d| < 0.33$ for small effects, $0.33 \leq |d| < 0.474$ for medium effects, and $|d| \geq 0.474$ for large effects (*Romano et al., 2006*). The python package used to compute Cliff's delta values can be accessed at https://github.com/neilernst/cliffsDelta (copy archived at *Ernst, 2021*). The input values used to compute Cliff's delta are log-transformed normalized gene expression values plus one as described in the 'Gene expression datasets' section of methods.

The DESeq2 LFC is computed as described above. To allow integration of the Cliff's delta effect size with the DESeq2 LFC, we also separated the LFC values into four levels: $|LFC| < log_2(1.1)$ for negligible effects, $log_2(1.1) \leq |LFC| < log_2(1.5)$ for small effects, $log_2(1.5) \leq |LFC| < log_2(2)$ or medium effects, and $|LFC| \geq log_2(2)$ for large effects.

For each gene, *g*, whether the gene is up-regulated or down-regulated in ecDNA(+) samples is determined by the signage and magnitude of its effect sizes $d_g$ and $LFC_g$. The initial criteria for $d_g$ and $LFC_g$ to be used as a determinant in the direction of a gene is for it to have a magnitude larger than that of a negligible effect. If the signage of $d_g$ and $LFC_g$ are both positive, gene *g* is considered up-regulated in ecDNA(+). If only a single value has a magnitude larger than the negligible effect threshold (e.g, $d_g > 0$ and $LFC_g = -0.1$), gene *g* is considered up-regulated in ecDNA(+). In the case of conflicting signages between the two values, the effect size with a larger magnitude takes precedence. For example, if $d_g = -0.2$ and $LFC_g = 0.847$, given that $d_g$ has a small negative effect and $LFC_g$ has a large positive effect, gene *g* is considered up-regulated in ecDNA(+) samples.

## Tumor heatmap

A Cliff's delta effect size matrix representing 643 CorEx genes was generated to compare TCGA with tumor expression patterns. For each of the 11 tumor types with at least 10 ecDNA(+) and at least 10 ecDNA(-) samples, we re-computed Cliff's delta. Using a Fisher's exact test (fisher_exact function from the scipy.stats python package; alternative hypothesis: two-sided), we tested the null-hypothesis of whether the up- and down- directionality of CorEx genes in TCGA vs. each tumor were independent of each other. The contingency table is as below. The directionality of a gene (up or down) was based solely on the signage of the gene's Cliff's delta effect size value.

|  |  | Tumor | |
| --- | --- | --- | --- |
|  |  | UP | DOWN |
|  | UP | a | b |
| TCGA | DOWN | c | d |

## Gene ontology (GO) enrichment analysis

To identify Gene Ontology Biological Process (GOBP) terms that were enriched in either the set of down-regulated or up-regulated CorEx genes, we applied one-sided Fisher's exact tests (alternative='greater;' scipy.stats python package) on 2x2 contingency tables for each GOBP term. Specifically, in the contingency table below, N is the total number of genes in the universe (i.e. 16 k for the number of genes measured in the RNAseq data), n is the number of DE genes (either up- or down-regulated in ecDNA(+) samples), m is the number of genes belonging to the GOBP term as defined by gene sets from MSigDB (c5.go.bp.v7.5.1.entrez), and k is the number of DE genes that belong to the GOBP term. The false discovery rate was controlled at 5% and adjusted $p$-values were computed using the Benjamini-Hochberg procedure (fdr correction from python statsmodels package). A final set of GOBP terms with adjusted $p$-value <0.05 was used for downstream analysis.

|  | DE | Non-DE |
| --- | --- | --- |
| Inside GOBP term | k | m-k |
| Outside GOBP term | n-k | N+k-n-m |

## Clustering gene sets

To cluster enriched gene sets into categories for visualization purposes, Cohen's kappa coefficient (python sklearn cohen_kappa_score) was used to determine term-term 'connectivity' (agreement of term-term pairs) – an approach described in the DAVID paper (*Huang et al., 2007*). Given a $r \times c$ binary matrix, where enriched GOBP terms are rows, CorEx genes are columns, and values are 1 if a CorEx gene is part of the GOBP term or 0 otherwise, kappa scores were computed between each pair of terms, where term $t_i \in t$ and $i = \{1, 2, 3, \ldots, n\}$: $Kappa\_Score\left(t_x, t_y\right)$, where $x \in i$ and $y \in i$.

Each term, $t_i$, formed an initial seeding group, $g_i$, where a term $t_x$ is part of $g_i$ if $Kappa\_Score\left(t_i, t_x\right) \geq score\ threshold$. If at least 50% of term-term pairs in $g_i$ have a $Kappa\_Score \geq score\ threshold$, the initial seeding group $g_i$ is retained for the next step. The second criteria ensures that terms within the same seeding group have strong interconnectivity. An iterative merging of seeding groups then follows: groups sharing p% or more members are merged. The representative term for each group was determined as the member with the highest interconnectivity score with other members of the group. After the automatic grouping process, a manual inspection leads to the merging of outliers or smaller groups into representative groups. We used a kappa score threshold of 0.5, and a condition of ≥25% of shared members when merging for the down-regulated genes, and a kappa score threshold of 0.6 and a condition of p≥50% of shared members when merging for the up-regulated genes.

## DDR pathway genes

We hand-curated 88 genes for double-stranded break DNA damage repair pathways (a-EJ, HR, c-NHEJ, SSA) via an extensive literature search, and added an additional 51 genes from the following MSigDB (c5.go.bp.v2022.1.Hs.entrez.gmt) GO biological process terms: GO:0097680 (double-strand break repair via classical nonhomologous end joining), GO:0097681 (double-strand break repair via alternative nonhomologous end joining), GO:1905168 (positive regulation of double-strand break repair via homologous recombination), and GO:0045002 (double strand break repair via single-strand annealing). This resulted in a final list of 129 genes. The directionality of the genes, as either up- or down-regulated in ecDNA(+) samples, is based on the full set of 1440 samples, consisting of 243 ecDNA(+) and 1197 ecDNA(-) samples representing 24 tumor types (*Supplementary file 1M*; *Supplementary file 1N*).

To test whether the number of genes passing our effect size thresholds for all genes 5256 (UP) and 5251 (DOWN) was significantly different from the up-/down-regulated genes implicated in each of the four pathways, we performed a Fisher's exact test (fisher_exact function from the scipy.stats python package) on the contingency table below, where $c$ and $d$ are the up- and down-regulated genes for each of the pathways tested.

Contingency table:

|  | UP | DOWN |
|---|---|---|
| All genes | 5256 | 5251 |
| Pathway | *c* | *d* |

Pathway values:

|  | c | d |
|---|---|---|
| c-NHEJ | 14 | 7 |
| Alt-EJ | 11 | 1 |
| HR | 46 | 8 |
| SSA | 11 | 1 |

## Physical presence on amplicons

To determine the physical presence of a gene on an amplicon, gene coordinates listed in the gene annotations file GRCh37/human_hg19_september_2011/Genes_July_2010_hg19.gff, downloaded from the AA repo on 3/21/2022, were mapped to amplicon genomic intervals (bed files). A gene is determined to be physically present on an amplicon if its genomic coordinates are fully encompassed within the amplicon genomic intervals.

## Mutational analysis

We pulled the list of mutations from the open-access version of the MC3 dataset (https://ellrottlab. org/project/mc3/), and then investigated differences between ecDNA(+) and ecDNA(-) samples. Synonymous mutations were excluded when calculating the mutation burden. For the differentially mutated gene analysis, only damaging mutations were selected by using snpEff (*Cingolani et al., 2012*) annotation which was originally included in the MC3 dataset. First, mutations annotated as HIGH in the IMPACT column were selected to obtain frameshift INDELs and stop gain SNVs. Next, mutations predicted to be damaging by SIFT (*Ng and Henikoff, 2003*) and PolyPhen2 (*Adzhubei et al., 2013*), were selected to obtain damaging missense mutations. Finally, we generated 2-by-2 contingency tables for each gene with cells *a*, *b*, *c*, and *d* representing the number of individuals with and without damaging mutations in ecDNA(+) and (-) tumors. The odds ratios were computed as $OR = \frac{ad}{bc}$, where *a* is the number of individuals in ecDNA(+) with the mutation, *b* is the number of individuals in ecDNA(+) without the mutation, *c* is the number of individuals in ecDNA(-) with the mutation, and *d* is the number of individuals in ecDNA(-) without the mutation. To determine if a gene contained mutations that were implicated in cancer, we checked genes against the Cancer Gene Census (CGC) database (v97) (*Sondka et al., 2018*), marking genes in the database as CGC and those that were not as non-CGC.

## Classification with mutational status

First, we created a binary matrix representing whether a gene is damaged or not, from the MC3 damaging mutation set described as above. Then we divided the whole matrix into 80% of the training set and 20% of the test set. Hyperopt was applied to the training set to select the best parameters for the XGBoost (*Chen and Guestrin, 2016*) model. The optimal parameters estimated by Hyperopt (eta = 0.1, max_depth = 6, min_child_weight = 3.0, scale_pos_weight = 4.9) were input to the XGBoost model, and the ecDNA status of each sample were also input as an answer set. The performance of the model was checked by inputting the 20% test set to the model and comparing the output result with the answer. Principal component analysis was also performed with the same mutational matrix as above, using the scikit.learn package.

## Ranking of CorEx genes

To rank CorEx genes by importance, we computed harmonic mean rank values based on three categories: (a) the average Gini importance statistic or mean decrease impurity (MDI) MDI (feature importance) values extracted from the trained random forest models on 200 training sets during the evaluation method described above, (b) the number of Boruta trials (out of 200) that a gene is

selected in, rounded to 2-digits, and (c) the number of Boruta trials (out of 200) that a gene is selected in when counting by cluster, rounded to 2-digits. The ranks for each category are adjusted separately so that genes with the same value share the same rank value. For example, if using the Boruta trial count as the rank value:

| | Trial count | Round (Trial count /10.0) | Rank | Adjusted rank |
|---|---|---|---|---|
| Gene A | 200 | 20 | 1 | 1 |
| Gene B | 200 | 20 | 2 | 1 |
| Gene C | 200 | 20 | 3 | 1 |
| Gene D | 187 | 19 | 4 | 4 |

The harmonic mean rank is defined as:

$$harmonic\ mean\ rank = \frac{3}{\frac{1}{a} + \frac{1}{b} + \frac{1}{c}}$$

## Tumor immune subtype

We classified ecDNA(+) and ecDNA(-) samples into the six immune subtype categories (*Thorsson et al., 2018*) provided in Table S6 from *Bagaev et al., 2021*.

## Impact of tumor purity on CorEx gene expression

To investigate the effects of the presence of non-cancer tissue (impurity) in bulk RNA-seq samples on the analyses performed in this study, we utilized the consensus measurement of purity estimations (CPE) for TCGA samples from a publication by *Aran et al., 2015*. Of the 870 TCGA samples (234 ecDNA(+), 636 ecDNA(-)) with gene expression (RNA-seq) data, 701 samples (174 ecDNA(+), 527 ecDNA(-)) were assigned a CPE value by Aran et al.,. To determine if the presence of undetected ecDNA in ecDNA(-) samples would confound the results by reducing the discriminating power of genes, we measured the expression directionality of CorEx genes in all samples (n=870) versus samples which had a high tumor purity (CPE ≥0.8, n=287). Specifically, p-values were obtained by performing Mann-Whitney U rank tests (scipy.stats python package) on gene expression values of ecDNA(+) and ecDNA(-) samples for both the 870 TCGA samples and 287 TCGA samples with high tumor purity. Genes with a significantly (p-value ≤0.05) higher expression in ecDNA(+) samples (alternative='greater') were labeled as 'UP,' while genes with a significantly lower expression in ecDNA(+) samples (alternative='less') were labeled as 'DOWN.' To generate a plot that compared the gene directionality of all samples vs. high-purity samples using p-values, a function $F$ was applied to p-values. Specifically, $F(p) = d \cdot log_{10}(p)$, where $p$ is the p-value and $d = 1$ if directionality is 'DOWN' and $d = -1$ if directionality is 'UP.'

## Additional information

### Competing interests

Jens Luebeck: J.L. receives compensation as a consultant for Boundless Bio. Howard Y Chang: Reviewing editor, eLife. Sihan Wu: S. Wu is a member of the scientific advisory board of Dimension Genomics Inc. Paul S Mischel: P.S.M. is a co-founder and advisor of Boundless Bio. J.L. receives compensation as a consultant for Boundless Bio. Vineet Bafna: V.B. is a co-founder, paid consultant, SAB member and has equity interest in Boundless Bio, Inc and Abterra Biosciences, Inc. The other authors declare that no competing interests exist.

### Funding

| Funder | Grant reference number | Author |
|---|---|---|
| Cancer Research UK | CGCATF-2021/100012 | Howard Y Chang Paul S Mischel |
| Cancer Research UK | CGCATF-2021/100025 | Vineet Bafna |

| Funder | Grant reference number | Author |
|---|---|---|
| National Cancer Institute | OT2CA278635 | Vineet Bafna |
| National Cancer Institute | OT2CA278688 | Howard Y Chang |
| National Institutes of Health | R01GM114362 | Miin S Lin<br>Jens Luebeck<br>Vineet Bafna |
| National Cancer Institute | U24CA264379 | Jens Luebeck<br>Vineet Bafna |
| Cancer Research UK | CGCATF-2021/100023 | Sihan Wu |
| National Cancer Institute | OT2CA278683 | Sihan Wu |
| Cancer Prevention and Research Institute of Texas | RR210034 | Sihan Wu |
| Korea Health Industry Development Institute | HI19C1330 | Se-Young Jo |

The funders had no role in study design, data collection and interpretation, or the decision to submit the work for publication.

### Author contributions

Miin S Lin, Conceptualization, Data curation, Formal analysis, Methodology, Writing – original draft, Writing – review and editing, All analyses were performed by M.S.L., apart from the following ones. Mutational analysis was performed by S.J. The amplicon classifications were performed by J.L; Se-Young Jo, Data curation, Formal analysis, Writing – original draft; Jens Luebeck, Data curation, Writing – original draft; Howard Y Chang, Sihan Wu, Supervision, Writing – review and editing; Paul S Mischel, Conceptualization, Supervision, Methodology, Writing – review and editing; Vineet Bafna, Conceptualization, Supervision, Funding acquisition, Methodology, Writing – original draft, Writing – review and editing

### Author ORCIDs

Miin S Lin ⓘ https://orcid.org/0000-0003-2017-4246
Howard Y Chang ⓘ https://orcid.org/0000-0002-9459-4393
Sihan Wu ⓘ https://orcid.org/0000-0001-8329-7492
Vineet Bafna ⓘ https://orcid.org/0000-0002-5810-6241

Reviewer #1 (Public Review): https://doi.org/10.7554/eLife.88895.3.sa1
Reviewer #2 (Public Review): https://doi.org/10.7554/eLife.88895.3.sa2
Reviewer #3 (Public Review): https://doi.org/10.7554/eLife.88895.3.sa3
Author response https://doi.org/10.7554/eLife.88895.3.sa4

## Additional files

### Supplementary files

• Supplementary file 1. Supplementary Tables. (**A**) CorEx genes. (**B**) Extrachromosomal DNA (ecDNA) status of 870 The Cancer Genome Atlas (TCGA) samples across 14 tumor types. (**C**) Highly co-expressed gene clusters identified using multiscale bootstrap resampling. (**D**) Cluster #3 and #74 members. (**E**) Cliff's delta values of 643 CorEx genes in TCGA samples, and in 11 tumor types with at least 10 ecDNA(+) and 10 ecDNA(-) samples each. (**F**) Top-|LFC| genes: 643 most significantly differentially expressed genes based on logarithmic fold changes from a DESeq2 analysis. (**G**) Up-/down-regulated genes in 870 TCGA samples. (**H**) Gene Ontology (GO) biological processes enriched in up-regulated CorEx genes. (**I**) Clustering of GO biological processes enriched in up-regulated CorEx genes into 11 broad categories. (**J**) GO biological processes enriched in Cluster #3 genes. (**K**) Up-regulated CorEx genes unique to biological process categories. (**L**) Hand-curated list of 129 genes involved in DSB repair pathways. (**M**) ecDNA status of 1440 TCGA samples across 24 tumor types. (**N**) Up-/down-regulated genes in 1,440 TCGA samples. (**O**) GO biological processes enriched in down-regulated CorEx genes. (**P**) Clustering of GO biological processes enriched in

down-regulated CorEx genes into seven broad categories. (**Q**) Down-regulated CorEx genes in NF-κB signaling (GO:0007249) involved in pro-apoptotic caspase activation. (**R**) Differentially mutated genes in ecDNA(+) and ecDNA(-) samples and their odds ratios.

• MDAR checklist

## Data availability

Public data from cBioPortal and Broad Firehose were used for this study. Scripts to generate CorEx genes are located on GitHub at https://github.com/miinslin/ecDNA_Gene_Expression (copy archived at *Lin, 2024*).

The following previously published datasets were used:

| Author(s) | Year | Dataset title | Dataset URL | Database and Identifier |
|---|---|---|---|---|
| Akbani R | 2015 | EB++AdjustPANCAN_IlluminaHiSeq_RNASeqV2.geneExp.tsv | https://www.synapse.org/#!Synapse:syn4976363 | synapse, syn4976363 |
| TCGA | 2018 | mRNAseq_Preprocess.Level_3 | https://gdac.broadinstitute.org/ | Broad Firehose, *.uncv2.mRNAseq_raw_counts.txt |

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
