## [Editor Report · eLife assessment]

This study of extrachromosomal DNA (ecDNA) identifies genes that distinguish ecDNA+ and ecDNA- tumors. The findings in the manuscript are **important** and the genomic analyses **convincing**. However, some of the data remain observational and the inferences would therefore be more robust with experimental validation. This manuscript could well be of relevance to biologists interested in cancer biology and gene regulation.

---

## [Referee Report · Reviewer #1 (Public Review)]

Recently discovered extrachromosomal DNA (ecDNA) provides an alternative non-chromosomal means for oncogene amplification and a potent substrate for selective evolution of tumors. The current work aims to identify key genes whose expression distinguishes ecDNA+ and ecDNA- tumors and the associated processes to shed light on the biological mechanisms underlying ecDNA genesis and their oncogenic effects. This is clearly an important question and through detailed analysis this work points to specific GO processes associated (up and down) with ecDNA+ tumors, namely, specific DNA damage repair processes and specific oncogenic processes.

In the initial submission I had commented on lack of clarity of method, potential biases, and in some cases inappropriate interpretation. In the revised version, the authors have addressed all my comments satisfactorily and I think this is an important work furthering our understanding of mechanisms underlying ecDNA+ tumors.

---

## [Referee Report · Reviewer #2 (Public Review)]

In their manuscript Lin et al. describe an important study on the transcriptional programs associated with the presence of extrachromosomal DNA in a cohort of 870 cancers of different origins. The authors find that compared to cancers lacking such amplifications, ecDNA+ cancers express higher levels of DNA damage repair-associated genes, but lower levels of immune-related gene programs.

This work is very timely and its findings have the potential to be very impactful, as the transcriptional context differences between ecDNA+ and ecDNA- cancers are currently largely unknown. The observation that immune programs are downregulated in ecDNA+ cancers may initiate new preclinical and translational studies that impact the way ecDNA+ cancers are treated in the future. Thus, this study has important theoretical implications that have the potential to substantially advance our understanding of ecDNA+ cancers.

Strengths:

The authors provide compelling evidence for their conclusions based on large patient datasets. The methods they used and analyses are rigorous.

Weaknesses:

The biological interpretation of the data remains observational. The direct implication of these genes in ecDNA(+) tumors is not tested experimentally.

---

## [Referee Report · Reviewer #3 (Public Review)]

Summary:

Using a combination of approaches, including automated feature selection and hierarchical clustering, the author identified a set of genes persistently associated with extrachromosomal DNA (ecDNA) presence across cancer types. The authors further validated the gene set identified using gene ontology enrichment analysis and identified that upregulated genes in extrachromosomal DNA-containing tumors are enriched in biological processes like DNA damage and cell proliferation, whereas downregulated genes are enriched in immune response processes.

Comments for the previous version:

Major comments:

(1) The authors presented a solid comparative analysis of ecDNA-containing and ecDNA-free tumors. An established automated feature selection approach, Boruta, was used to select differentially expressed genes (DEG) in ecDNA(+) and ecDNA(-) TCGA tumor samples, and the iterative selection process and two-tier multiple hypothesis testing ensured the selection of reliable DEGs. The author showed that the DEG selected using Boruta has stronger predictive power than genes with top log-fold changes.

(2) The author performed a thorough interpretation of the findings with GO enrichment analysis of biological processes enriched in the identified DEG set and presented interesting findings, including the enrichment in DNA damage process among the genes upregulated in ecDNA(+) tumors.

(3) Overall, the authors achieved their aims with solid data mining and analysis approaches applied to public data tumor data sets.

(4) While it may not be the scope of this study, it will be interesting to at least have some justification for choosing Boruta over other feature selection methods, such as Recursive Feature Elimination (RFE) and backward stepwise selection.

(5) The authors showed that DESEQ-selected DEGs with top log-fold changes have less strong predictive power and speculated that this may be due to the fact that genes with top log-fold changes (LFC) are confined only to a small subset of samples. It will be interesting to select DEGs with top log-fold changes after first partitioning the tumor samples. For example, randomly partition the tumor samples, identify the DEGs with top LFC, combine the DEGs identified from each partition, then evaluate the predictive power of these DEGs against the Boruta-selected DEGs.

(6) While the authors showed that the presence of mutations was not able to classify ecDNA(+) and (-) tumor samples, it will be interesting to see if variant allele frequencies of the genes containing these mutations have predictive power.

Comments for the revised version:

The authors addressed the comments and recommendations with solid analysis and explanations in the revision. The added analysis using GLM is especially appreciated and provides convincing evidence for the predicting power of the Boruta-selected genes. The only comment is at this point is that it is recommended that the author provide some justification for choosing Boruta over other feature selection methods. It is not necessary to provide benchmarking results - justification based on the review of previous literature is sufficient, as it is not well explained in the paper why Boruta was chosen in the first place. Is it state-of-the-art? Has it demonstrated better performance in other settings? A few sentences answering these questions should suffice.

---

## [Author Response]

The following is the authors’ response to the original reviews.

**eLife assessment**
This study of extrachromosomal DNA (ecDNA) aims to identify genes that distinguish ecDNA+ and ecDNA- tumors. This timely study is important in addressing the genes responding to the amplification of the ecDNA. The data presented are for the most part solid, there were concerns regarding the clarity in the description of the analysis methods and whether the evidence for specific genes required to maintain the ecDNA+ state was entirely conclusive.
**Public Reviews:**

**Reviewer #1 (Public Review):**
Recently discovered extrachromosomal DNA (ecDNA) provides an alternative non-chromosomal means for oncogene amplification and a potent substrate for selective evolution of tumors. The current work aims to identify key genes whose expression distinguishes ecDNA+ and ecDNA- tumors and the associated processes to shed light on the biological mechanisms underlying ecDNA genesis and their oncogenic effects. While this is clearly an important question, the analysis and the evidence supporting the claims are weak. The specific machine learning approach seems unnecessarily convoluted, insufficiently justified and explained, and the language used by the authors conflates correlation with causality. This work points to specific GO processes associated (up and down) with ecDNA+ tumors, many of which are expected but some seem intriguing, such as association with DSB pathways. My specific comments are listed below.

**Response.** As some of the specific questions below address similar concerns, we have answered them briefly here. As a high level point, the reviewer is correct in that other statistical or ML approaches could potentially have been used, and that some are simpler. However, the test used here directly addresses the question: *Find a collection of genes whose expression value is predictive of ecDNA status in the sample.* Because the underlying method in the Boruta analysis uses random forests, it can test predictive power without relying on a linearity assumption implicit in other methods. In this revision, we also compare against a Generalized Linear Model and show that it is less suited to the specific task above. We also address the reviewer concerns about specific parameter choices by showing robustness to the specific parameter.

(A) The claim of identifying genes required to 'maintain' ecDNA+ status is not justified - predictive features are not necessarily causal.

**Response.** We agree with the reviewer that predictive features are correlative and not causal. In the manuscript, we identify genes whose expression (when used as a feature) is predictive of ecDNA presence or absence. Such predictive genes are consistently over-expressed or consistently under-expressed in ecDNA(+) samples relative to ecDNA(-) samples even though they are not required to be on ecDNA. To our knowledge, we did not claim that these genes are causal for ecDNA formation or maintenance, only that such genes and the underlying biological processes are worth investigating. In the beginning of the manuscript, we had written the following paragraph, but we have removed the last line:

“In lieu of identifying genes that are highly differentially expressed between ecDNA(+) and ecDNA(-) samples but driven by a small subset of cases (e.g. gene A in Figure 1—figure supplement 1), we sought to identify genes (e.g. gene B) whose expression level was predictive of ecDNA presence. We assumed that genes that were persistently over-expressed or under-expressed in ecDNA(+) samples relative to ecDNA(-) samples were more likely to be involved in ecDNA biogenesis or maintenance, or in mediating the cellular response to the presence of ecDNA.”

We revised the manuscript to make sure that there are no claims that refer to causality. We revisited all phrases where the words like “maintain” were used and added appropriate disclaimers, or replaced them by the phrase, “ecDNA presence.” The remaining statements say, for example, “These results are consistent with a pan-cancer role of CorEx genes in ecDNA biogenesis and maintenance,” and do not claim causality.

(B) The methods and procedures to identify the key genes is hyper-parameterized and convoluted and casts doubt on the robustness of the findings given the size and heterogeneity of the data.(a) In the first two paragraphs of Boruta Analysis Methods section, authors describe an iterative procedure where in each iteration, a binomial p-value is computed for each gene based on number of iterations thus far in which the gene was selected (higher GINI index than max of shadow features). But then in the third paragraph they simply perform Random Forest in 200 random 80% of samples and pick a gene if it is selected in at least 10/200. It is ultimately not clear what was done. Why 10/200? Also "the probability that a gene is a "hit" or "non-hit" in each iteration is 0.5" is unclear. That probability is of a gene achieving GINI index higher than the max of shadow features. How can it be 0.5?

**Response.** We believe that there is some misunderstanding about the algorithm, and we agree that the description should have been more clear. We have greatly simplified the description in the manuscript. However, we want to provide some higher-level explanation here. Boruta is a standard feature extraction algorithm (Kursa, *Journal of Statistical Software* September 2010, Volume 36, Issue 11), and we used a Python implementation of the method. Given a gene expression data-set with class labels on samples, Boruta extracts features (genes) that best predict the class labels using a Random Forest Classifier, as long as the features are more predictive than permuted features added in each iteration. As we are using an implementation of a published method, we have removed non-essential details, referring directly to the publication. Nevertheless, to address the reviewer’s specific critique, the number of false-features added changes in each iteration (it equals the number of accepted+uncommitted features). Therefore, the choice of 0.5 by Boruta (it is fixed in the published method and not a user-specified parameter) is a conservative approach. If a gene was no better than a randomly chosen feature, its predictive performance would exceed the *most predictive* randomly chosen feature by at most 0.5 (but could be lower, making the choice of 0.5 conservative).

While Boruta iteratively picks genes that are significantly better than random features, the list of genes predicted might be specific to the data-set, and might change with different data-sets. Therefore, we employed a bootstrapping strategy: we performed 200 trials each time picking 80% of the ecDNA(+) samples and 80% of the ecDNA(-) samples at random, thus generating many data-sets while maintaining class imbalance. For each of the 200 trials, we performed a Boruta analysis. Finally, we picked a gene if it was selected as a Boruta feature in at least 10 of 200 trials.

The reviewer has a reasonable critique about why 10 (of 200) specifically, and why not fewer or more. Most genes are weak predictors by themselves. For example, RAE1, which is the top ranked gene, picked in all 200 Boruta trials, can only predict ecDNA status with poor recall for any meaningful precision.

**Author response image 1. sa4fig1:** 

Given the weakness of an individual gene as a classifier, its repeated selection in multiple Boruta trials is already a significant event. By requiring a gene to be picked in 5% of the trials (10/200), we were selecting a small, but more robust list of genes. However, to further explore the reviewer’s concerns, we also applied 8 other selection criteria ranging from 5 (of 200 Boruta trials) to 200 of 200 Boruta trials. See Figure below. The number of CorEx genes expectedly decreases. However, of the 187 GO terms that were enriched by 262 UP-genes using 10 of 200 Boruta trials as the selection criteria, 93 terms (49.7%) were enriched for each cut-off (see Author response image 2), and 155 terms (82.9%) were enriched in at least 5 of the 8 cut-off criteria. Given that the remaining analysis works on the hierarchy of GO terms and finds 4 GO-categories (Mitotic Cell Cycle, G1/S, G2/M; cell-division; DSB DNA Damage response; and the HOX Gene cluster) enriched by UP-regulated genes, those conclusions would hold regardless of the specific cut-off.

**Author response image 2. sa4fig2:** 

The number of GO terms that were enriched by DOWN-regulated genes is smaller, only 73, and falls rapidly for higher cut-offs, with 25 at a cut-off of 15. Therefore we see fewer terms enriched for more stringent cut-offs. However, they all support immune processes. These results do suggest that there are fewer genes that are consistently down-regulated in ecDNA(+) cancers, and expression change in a small number of genes may be sufficient to promote conditions for ecDNA.

Finally, we note that in the final section we discuss the 65 most highly ranked genes with a harmonic mean rank <= 3. These 65 CorEx genes (or a member of their cluster) appear in each of 200 Boruta trials. Thus, their choice is also not dependent on the cut-off of 10 in 200. In summary, the conclusions of the paper do not depend upon the specific cut-off of 10 in 200 trials.

We have added the figure as a supplemental figure and have added the following text to the manuscript:

“Any CorEx gene is either a Core gene that was selected as a feature in at least 5% of 200 Boruta trials, or be highly co-expressed with a Core gene. Because the selection criterion of 5% is arbitrary, we also tested robustness with 8 other cut-offs ranging from 5-of-200 to 200-of-200 Boruta trials. The number of CorEx genes expectedly decreases with more stringent cut-offs. However, of the 187 GO terms that were enriched by 262 CorEx UP-genes using 10 of 200 Boruta trials as the selection criteria, 93 terms (49.7%) were enriched for each cut-off (Figure 1—figure supplement 5), and 155 terms (82.9%) were enriched in at least 5 of the 8 cut-offs. Given that our subsequent analyses utilized the hierarchy of GO terms and identified 4 GO-categories enriched by UP-regulated genes, the conclusions would hold regardless of the specific cut-off.”

(b) The approach of combining genes with clusters is arbitrary. Why not start with clusters and evaluate each cluster (using some gene set summary score) for their ability to discriminate? Ultimately, one needs additional information to disambiguate correlated genes (i.e. in a coexpression cluster) in terms of causality.

**Response.** In general, the approach proposed by the reviewer is reasonable. However, we did consider that possibility and found that our approach was easier to implement. For example, if we clustered first, we would have the challenge of choosing the correct set of clusters. Also, the Boruta analysis would become very difficult while dealing with clusters (e.g., how to define falsefeatures?). We tested other methods of picking genes that were suggested by other reviewers such as generalized linear models. They turned out not to be as predictive of ecDNA status, as described later in the response. Finally, we performed many experiments to ensure the validity of the clustering. Specifically, we had the following text in the paper:

“Notably, among the 354 clusters, only 2 clusters (with 14 total genes) did not contain any Core genes. As most genes do not have completely identical expression patterns, we would expect one gene to be consistently picked as a Boruta gene over another co-expressed gene. Consistent with this hypothesis, most (344/354) clusters contained only 1 or 2 Core genes (Figure 1). When selecting clusters that contained at least 1 Core and 1 co-expressed gene, 53 of 71 clusters contained 1 to 3 Core genes (Figure 1—figure supplement 2), confirming that a few genes per co-expressed cluster provide sufficient predictive value, but other co-expressed genes might still play an important functional role in maintaining ecDNA(+) status.”

These experiments suggest that the genes found by extending the Core genes through clustering do not radically change the Core genes, but only enhance the set.

(c) The cross-validation procedure is not clear at all. There is a mention of 80-20 split but exactly how/if the evaluation is done on the 20% is muddled. The way precision-recall procedure is also a bit convoluted - why not simply use the area under the PR curve?

**Response.** We apologize if the method was unclear. We have rewritten the methods part to make things clearer. As a high level point, there are *two places* where we use the same 80-20 split, and that resulted in some confusion. We start by randomly picking 80% of the ecDNA(+) and 80% of ecDNA(-) samples to create an 80-20 split of all samples. This procedure is repeated to generate 200 80-20 split data-sets. These data-sets are hereafter called 200 training and test samples.

In the first usage, we use only the ‘training’ part of the 200 samples. We apply Boruta to each training set, and this helps us select the Core genes, which are then expanded to form the CorEx set. At this point, the CorEx genes are frozen for analysis in the rest of the paper. One question that we subsequently answer is *what is the predictive power of the CorEx genes in determining if the sample is ecDNA(+) or ecDNA(-)?* We also compare the predictive performance of CorEx genes relative to (a) Core genes, (b) LFC genes, and (c) random genes. In the revised manuscript, we have added another list of 3,012 genes selected using a single gene generalized linear model (GLM) for feature prediction. To make these comparisons, we utilized the same 200 training and test data-sets as before. In each test, we trained a random forest classifier on the training set and predicted on the ‘test’ set, for each of the 5 gene lists. This provided a uniform and fair method for testing which of the 5 gene lists was the better predictor of ecDNA status.

The precision recall values are plotted in Figure 2 (also included below). We note that *none of the* gene lists was a great predictor of ecDNA status of a sample. However, the CorEx and Core genes were significantly more predictive than GLM, LFC, and random genes. The predictive power of GLM genes was very similar to LFC, and better than random.

For each of these 200 tests, we obtained a separate area under the precision-recall curve number for each of the gene-sets. To address the reviewer’s comments regarding a single number, we reported the average of the AUPRC for each of the gene-sets in the revision. The mean AUPRC values were added to the manuscript and are described here as well: Core_408_genes (0.495), CorEx_643_genes (0.48), Random_643_genes (0.36), top_lfc_643_genes (0.429), and GLM_R_3012_genes (0.426).

We also changed Figure 2 to show box-plots showing distribution of recall values for specific precision windows instead of maximum recall. For ease of checking, the figure is reproduced below.

**Author response image 3. sa4fig3:** 

(d) The claim is that Boruta genes are different from differentially expressed genes but the differential expression seems to be estimated without regards to cancer type, which would certainly be highly biased and misleading. Why not do a simple regression of gene expression by ecDNA status, cancer type and select the genes that show significant coefficient for ecDNA status?

**Response.** As requested by the reviewer, and in the more detailed questions below, we added an alternative model with a generalized linear model (GLM) analysis that controlled for tumor subtype. The method itself is described in the Methods section and pasted below. The GLM genes were tested along with the LFC, CorEx, Core genes as described in response to the previous question, and those results are now presented in Figure 2 and the revised manuscript.

“We tested each of 16,309 genes independently in a separate logistic regression model using the glm() function in the R stats package (v4.2.0), and retained genes that were significant(*p*-value 0.01). Specifically, the model was defined as glm(𝑦 ~ 𝑔_𝑗_ + 𝑡, data = 𝑀, family = binomial(link = 'logit')), where 𝑦 is the response vector where 𝑦_𝑖_=1 if sample 𝑖 ∈ {1, . . . ,870} is ecDNA(+) and 𝑦_𝑖_ =0 otherwise, 𝑔_𝑗_ is the vector of expression values for gene *j* ∈ {1, . . . ,16309} in samples 𝑖 ∈ {1,. . . ,870}, *t* is the covariate vector representing the tumor subtypes of samples 𝑖 ∈ {1, . . . ,870}, and 𝑀 is the data matrix containing values of gene expression, tumor subtype, and ecDNA status for all samples. The equation for the binomial logistic regression described above is formulated as log⁡(p1−p)=β0+β1X1+…+βkXk where *p* is the probability that the dependent variable 𝑦 is 1, 𝑋 are the independent variables, and 𝛽 are the coefficients of the model. In this case, *k*=1 represents independent variable gene *j* and *k*=2 represents the tumor subtype covariate *t*. Of the 16,309 genes tested independently, 3,012 genes were significant at *p-*value<0.01.”

(C) After identifying key features (which the authors inappropriate imply to be causal) they perform a series of enrichment/correlative analysis.

Response. We have reviewed the document to ensure that we did not use the word ‘causal.’ If the reviewer can point to specific text, we are happy to change the phrasing.

(a) It is known that ecDNA status associates with poor survival, and so are cell cycle related signal. Then the association between Boruta genes and those processes is entirely expected. Is it not? The same goes for downregulation of immune processes.

**Response.** We agree with the reviewer that cell cycle related signals and immune related signals are associated with low survival, and so does ecDNA. However, many cellular processes could be associated with low survival (including for example, metabolic processes, protein and DNA biosynthesis, etc.). The unexpected part is that there appear to be only 4 major processes that are upregulated in ecDNA(+) cancers relative to ecDNA(-) cancers, and only one (immune response) that is downregulated.

(b) The association with DSB specifically is interesting. Further analysis or discussion of why this should be would strengthen the work.

**Response.** We thank the reviewer for their comment, and agree with their perspective. Note that we devoted a fair amount of text to analysis of DSB pathways. Specifically, we parsed the 4 main pathways in Figure 3, and found our data to suggest that many genes in the classical nonhomologous end joining repair pathway are down-regulated in ecDNA(+) samples relative to ecDNA(-) samples. In contrast, Alternative end-joining and homology directed repair pathways are upregulated. This is a surprising result because c-NHEJ is considered to be an important mechanism of DSB repair. We have some lines in the discussion that address this:

“The DNA damage genes are broadly up-regulated in ecDNA(+) samples, especially in double-strand break repair. Within this broad category of mechanisms, our analysis suggests that alternative DSB repair pathways such as Alt-EJ are preferred relative to classical NHEJ. This is consistent with previous observations of small microhomologies at breakpoint junctions, and has important implications in therapeutic selection that will need to be validated in future experimental studies. We note, however, the microhomology analyses typically study breakpoint junctions, and might ignore double-strand breaks in non-junctional sequences which could be observed, for example at replication-transcription junctions.”

We note that additional experimental work to corroborate these findings is significant effort and will be part of ongoing research in our collaborators’ laboratories.

(c) On page 15, second paragraph, when providing the up versus down CorEx genes, please also provide up versus down for non-CorEx genes as well to get a sense of magnitude.

**Response.** We thank the reviewer for the comment. We note that Supplementary file 1 has the complete contingency tables as well as the Fisher Exact Test statistic for all categories. For the specific categories mentioned in the paper, the chi-square tables are reproduced below. As we are citing Supplementary file 1 (containing all numbers and the statistic *p*-value) in the main text, we thought it was better to leave the text as it was.

Category: **Inflammation** (*p*-value: 0.005)

CorEx: 18 (UP), 76 (DOWN)

Non-CorEx: 325 (UP), 657 (DOWN)

Category: **Leukocyte migration and chemotaxis** (*p*-value: 0.03)

CorEx: 13 (UP), 49 (DOWN)

Non-CorEx: 213 (UP), 410 (DOWN)

Category: **Lymphocyte activation** (*p*-value: 0.0075)

CorEx: 23 (UP), 75 (DOWN)

Non-CorEx: 334 (UP), 560 (DOWN)

Category: **Cytokine production** (*p*-value: 0.117)

CorEx: 6 (UP), 28 (DOWN)

Non-CorEx: 93 (UP), 208 (DOWN)

(d) The finding that Boruta genes are associated with high mutation burden is intriguing because in general mutation burden is associated with better survival and immunotherapy response. This counter-intuitive result should be scrutinized more to strengthen the work.

**Response.** We agree with the reviewer that it is an intriguing observation. However, we are cautious in our interpretation. This is for the following reasons (all mentioned in the text):

(1) The total mutation burden was significantly higher in ecDNA(+) samples relative to ecDNA(-) samples (Figure 5). However, when controlling for cancer type, only glioblastoma, low-grade gliomas, and uterine corpus endometrial carcinoma continued to show differential total mutational burden (Figure 5—figure supplement 2).

(2) We tested if specific genes were differentially mutated between the two classes (Figure 5). For deleterious/high-impact mutations, TP53 was the only gene whose mutational patterns were significantly higher in ecDNA(+) compared to ecDNA(-) (OR 2.67, Bonferroni adjusted *p*-value 4.22e-07). BRAF mutations, however, were more common in ecDNA(-) samples and were significant to an adjusted *p*-value < 0.1 (OR 0.27).

(3) In response to another reviewer’s comment, we also tested correlation with variant allele frequencies, and did not find any significant correlation except for TP53. We decided not to include that result in the paper.

These tissue specific cases might be confounding the main observation, but we have placed all of them together so that the reader can gain a better understanding. It is worth noting that the correlation between high TMB and immunotherapy response is also now controversial, and perhaps not true for all cancer types. See for example (https://www.annalsofoncology.org/article/S0923-7534(21)00123-X/fulltext), which suggests that this relationship is not true for Glioma, and in Glioma (which is ecDNA enriched), higher TMB is associated with worse immunotherapy response. Our results are consistent with that finding. We have modified the discussion paragraph to better reflect this.

“Mutation data alone does not provide as clear a picture of the genes involved in ecDNA maintenance. We did observe that the total mutation burden (TMB) was higher in ecDNA(+) samples. However, that relationship is much less clear after controlling for cancer type. High TMB has been positively correlated with sensitivity to immunotherapy (47), and better patient outcomes; however, the gene expression patterns suggest that immunomodulatory genes are downregulated in ecDNA(+) samples, and patients with ecDNA(+) tumors have worse outcomes (28). Notably, other results have suggested that the correlation between TMB and response to immunotherapy is not uniform, and it can vary across different tumor subtypes (41). **Specifically, our data is consistent with previous results which showed that Gliomas with high TMB have worse response to immunotherapy relative to gliomas with low TMB** (41). In general, no collection of gene mutations was predictive of ecDNA status, although mutations in TP53 were more likely in ecDNA(+) samples, and perhaps are an important driver for ecDNA formation (38).”

(e) On page 17 "12 of the 47 genes not specifically enriching any known GO biological Process" is confusing. How can individual gene enrich for a GO process?

**Response.** We agree that the statement was incorrectly phrased. We have changed it to state that “Only 12 of the 47 genes were not included in the gene sets of any enriched GO term.”

**Reviewer #2 (Public Review):**
In their manuscript entitled "Transcriptional immune suppression and upregulation of double stranded DNA damage and repair repertoires in ecDNA-containing tumors" Lin et al. describe an important study on the transcriptional programs associated with the presence of extrachromosomal DNA in a cohort of 870 cancers of different origin. The authors find that compared to cancers lacking such amplifications, ecDNA+ cancers express higher levels of DNA damage repair-associated genes, but lower levels of immune-related gene programs.This work is very timely and its findings have the potential to be very impactful, as the transcriptional context differences between ecDNA+ and ecDNA- cancers are currently largely unknown. The observation that immune programs are downregulated in ecDNA+ cancers may initiate new preclinical and translational studies that impact the way ecDNA+ cancers are treated in the future. Thus, this study has important theoretical implications that have the potential to substantially advance our understanding of ecDNA+ cancers.StrengthsThe authors provide compelling evidence for their conclusions based on large patient datasets. The methods they used and analyses are rigorous.WeaknessesThe biological interpretation of the data remains observational. The direct implication of these genes in ecDNA(+) tumors is not tested experimentally.

**Response.** We agree with the reviewer that experimental tests would be ideal. Towards that, there are some challenges. The immune system genes cannot be tested in cell line models as they need a tumor microenvironment. Tests of DSB repair mechanisms and cell cycle control can be performed in cell-lines, but not with the TCGA samples which are not available. Some of our collaborators are actively working on these topics, but that extensive experimental work is beyond the scope of this paper.

**Reviewer #3 (Public Review):**
Summary:Using a combination of approaches, including automated feature selection and hierarchical clustering, the author identified a set of genes persistently associated with extrachromosomal DNA (ecDNA) presence across cancer types. The authors further validated the gene set identified using gene ontology enrichment analysis and identified that upregulated genes in extrachromosomal DNA-containing tumors are enriched in biological processes like DNA damage and cell proliferation, whereas downregulated genes are enriched in immune response processes.Major comments:(1) The authors presented a solid comparative analysis of ecDNA-containing and ecDNA-free tumors. An established automated feature selection approach, Boruta, was used to select differentially expressed genes (DEG) in ecDNA(+) and ecDNA(-) TCGA tumor samples, and the iterative selection process and two-tier multiple hypothesis testing ensured the selection of reliable DEGs. The author showed that the DEG selected using Boruta has stronger predictive power than genes with top log-fold changes.(2) The author performed a thorough interpretation of the findings with GO enrichment analysis of biological processes enriched in the identified DEG set, and presented interesting findings, including the enrichment in DNA damage process among the genes upregulated in ecDNA(+) tumors.(3) Overall, the authors achieved their aims with solid data mining and analysis approaches applied to public data tumor data sets.(4) While it may not be the scope of this study, it will be interesting to at least have some justification for choosing Boruta over other feature selection methods, such as Recursive Feature Elimination (RFE) and backward stepwise selection.

**Response.** We actually agree with the reviewer that some other feature selection methods could work just as well, and note that the Boruta analysis is not our creation, but a published feature selection method (Kursa, *Journal of Statistical Software* September 2010, Volume 36, Issue 11). We use Boruta to identify relevant genes, but the bulk of the paper is to understand the biological processes driven by that gene selection. Even if we had chosen another method that performed slightly better, it likely would not change the main conclusions. However, to address the reviewers concerns on over-reliance on one method, we added a different gene list created by a generalized linear model analysis, with the goal of checking if the expression of a gene could predict the ecDNA status of the sample *after controlling for tumor subtype*. Thus, we tested 5 different genelists in terms of their power in predicting ecDNA. While none of the lists is a great predictor of ecDNA status, the Core and CorEx gene lists are significantly better than the other lists. The Figure below replaces the previous Figure panels 2b and 2c.

**Author response image 4. sa4fig4:** 

(1) The authors showed that DESEQ-selected DEGs with top log-fold changes have less strong predictive power and speculated that this may be due to the fact that genes with top log-fold changes (LFC) are confined only to a small subset of samples. It will be interesting to select DEGs with top log-fold changes after first partitioning the tumor samples. For example, randomly partition the tumor samples, identify the DEGs with top LFC, combine the DEGs identified from each partition, then evaluate the predictive power of these DEGs against the Boruta-selected DEGs.

**Response.** This is a great comment. We added a generalized linear model test for selecting genes whose expression is predictive of ecDNA status. The GLM list described above uses a standard methodology (Analysis of Variance) controls for tumor type as a covariate, and its predictive performance is only slightly better than the Top-|LFC| genes, while improving over a random gene set.

(2) While the authors showed that the presence of mutations was not able to classify ecDNA(+) and (-) tumor samples, it will be interesting to see if variant allele frequencies of the genes containing these mutations have predictive power.

**Response.** This is a great suggestion. To address the reviewer’s question, we used allelic counts (REFs and ALTs) information from the MC3 variant callset, and calculated allele frequencies of all variants from samples where ecDNA status was available. Next, we conducted a Wilcoxon rank-sum test between VAFs of the ecDNA(+) group and VAFs of the ecDNA(-) group for every mutated gene. We found 1,073 genes with *p*<0.05, but among them, only TP53 passed the multiple testing correction (*padj*<0.05, Benjamini-Hochberg). As the results are identical to the tests based solely on presence of mutations, we decided not to include this data.

**Reviewer #1 (Recommendations For The Authors):**
(A) The presentation should be substantially streamlined.(B) Preferably use a more intuitive simpler ML approach with fewer parameters to make it more credible. Because there are relatively few samples across numerous cancer types with greater variability in representation, a simpler procedure with transparent controls will be more convincing.

**Response.** We accept the reviewer’s criticism in that other statistical or ML approaches could potentially have been used, and that some are simpler. However, the test used here directly addresses the question: *Find a collection of genes whose expression value is predictive of ecDNA status in the sample.* Because the underlying method in the Boruta analysis uses random forests, it can test predictive power without relying on a linearity assumption implicit in other methods. In this revision, we also compare against a Generalized Linear Model (regression analysis) and show that it is less suited to the specific task above. We address the reviewer concerns about specific parameter choices by showing robustness to the specific parameter. All details are provided in the initial questions, and in the revised manuscript.

(C) Avoid using any term implying causality unless you can bring in direct experimental evidence e.g. mutagenesis experiment followed by ecDNA measurement. Some places you use the word 'maintain ecDNA' and other places 'ecDNA impact'. But these are all associations. How can you distinguish causal genes from downstream effects without additional data?

**Response.** We note that the word causal does not appear anywhere in the manuscript, and was not intended. Additionally we have revised the manuscript and are open to specific changes requested by the reviewer or the editors.

(D) Along these lines, if Boruta genes are indeed causal, one would expect Boruta-Up genes to be amplified more than expected in the ecDNA+; converse for Boruta-down genes.

**Response.** We did not understand the reviewer’s question. By “amplified,” if the reviewer means “amplification of transcript level,” then that is exactly what the Boruta analysis is showing. Specifically, for each gene, we have the ability to pick a transcript level cut-off ‘*t*’ so that samples in which the expression is higher than *t* are more likely to be ecDNA(+). However, we are not claiming that there is causality, just that the transcript level is (weakly) predictive of the ecDNA status of the sample.

(E) A strawman control should be a simple regression-based gene identification that controls for ecDNA status and cancer type.

**Response.** We agree that this was a very good suggestion. In the revision, we have applied a GLM, which controls for tumor type. Thus, we have 5 gene-lists (including the Core and CorEx genes). As described in the revised manuscript but also in response to the main comments above, none of the lists are a great predictor. However, the CorEx and Core genes are significantly better at predicting ecDNA status of a sample.

**Reviewer #2 (Recommendations For The Authors):**
Comments(1) The analysis hinges on a classification of tumors into ecDNA(+) and ecDNA(-) using AmpliconClassifier. It would be good to know how robust the outcomes are with respect to the performance of AmpliconClassifier - how many false positives and negatives willAmpliconClassifier generate on this dataset and how would this influence the CorEx genes?

**Response.** This is a very reasonable request. AA has been extensively tested on established cell-lines for its ability in predicting ecDNA status, and this information is published in multiple venues, including Kim, Nature genetics 2020, and shows precision 85% for recall 83%. For completeness, we have reproduced the relevant plot from that paper here, and the relevant text here, but are not including it in the manuscript.

“To evaluate the accuracy of the AmpliconArchitect predictions, we analyzed whole-genome sequencing data from a panel of 44 cancer cell lines, and examined tumor cells in metaphase. We used 35 unique fluorescence in-situ hybridization (FISH) probes in combination with matched centromeric probes (81 distinct “cell-line, probe” combinations) to determine the intranuclear location of amplicons (Supplementary Table 2). Following automated analysis >1,600 images, we observed that 85% of amplicons characterized as ‘Circular’ by whole genome sequencing profile demonstrated an extrachromosomal fluorescent signal, representing the positive predictive value. Of the amplicons corresponding to extrachromosomally located FISH probes, 83% were classified as Circular, representing the sensitivity (Extended Data Fig. 1A).”

**Author response image 5. sa4fig5:** 

(2) It is unclear why genes are labeled Boruta genes when they are present in 10 out of 200 runs, this seems like an unexpectedly low number. How did the authors arrive at this number? Do the authors have any ground truth to estimate how well Boruta works in this setting and implementation?

**Response.** This is a great question and asked by another reviewer as well. Given the weakness of an individual gene as a classifier, its repeated selection in multiple Boruta trials is already a significant event. By requiring a gene to be picked in 5% of the trials (10/200), we were selecting a small, but more robust list of genes. However, to further explore the reviewer’s concerns, we also applied 8 other selection criteria ranging from 5 (of 200 Boruta trials) to 200 of 200 Boruta trials. See Figure below. The number of CorEx genes expectedly decreases with increasing stringency. However, of the 187 GO terms that were enriched by UP-genes, 93 terms (50%) were enriched regardless of the cut-off (see Figure below), and 153 terms (82%) were enriched in at least 5 of the 8 cut-offs. Given that the remaining analysis works on the hierarchy of GO terms and finds 4 GO-categories (Mitotic Cell Cycle, G1/S, G2/M; cell-division; DSB DNA Damage response; and the HOX Gene cluster) enriched by UP-regulated genes, those conclusions would hold regardless of the specific cut-off.

**Author response image 6. sa4fig6:** 

The number of GO terms that were enriched by DOWN-regulated genes is smaller, only 73, and falls rapidly for higher cut-offs, with 25 at a cut-off of 15. Therefore we see fewer terms enriched for more stringent cut-offs. However, they all support immune processes. These results do suggest that there are fewer genes that are consistently down-regulated in ecDNA(+) cancers, and expression change in a small number of genes may be sufficient to promote conditions for ecDNA.

We have added the figure as a supplemental figure and have added the following text to the manuscript:

“Any CorEx gene is either a Core gene that was selected as a feature in at least 5% of 200 Boruta trials, or be highly co-expressed with a Core gene. Because the selection criterion of 5% is arbitrary, we also tested robustness with 8 other cut-offs ranging from 5-of-200 to 200-of-200 Boruta trials. The number of CorEx genes expectedly decreases with more stringent cut-offs.

However, of the 187 GO terms that were enriched by 262 CorEx UP-genes using 10 of 200 Boruta trials as the selection criteria, 93 terms (49.7%) were enriched for each cut-off (Figure 1—figure supplement 5), and 155 terms (82.9%) were enriched in at least 5 of the 8 cut-offs. Given that our subsequent analyses utilized the hierarchy of GO terms and identified 4 GO-categories enriched by UP-regulated genes, the conclusions would hold regardless of the specific cut-off.”

(3) Authors extend the core gene set with co-expressed genes, arguing that "gene C" would not add predictive power in addition to "gene B" and is therefore not identified as a Boruta gene. However, from its description in the manuscript (summarized: "Boruta [...] selects the highest feature importance score, s, of shadow features as a cut off, and returns features with a higher score than s."), it isn't immediately obvious to me why Boruta would not return both genes B and C. Maybe the authors could explain this better.

**Response.** We consider the following.

(1) Consider 100 ecDNA(+) and 100 ecDNA(-) samples. Let the expression levels of genes B and C in the data-sets be as described in the figure below; y-axis is the gene expression, and x-axis is just a listing of all samples, with green color denoting ecDNA(+) samples and orange color denoting ecDNA(-) samples.

**Author response image 7. sa4fig7:** 

(2) Then, if we choose gene B and a transcript level of 1.25, we have a perfect prediction of ecDNA status because all samples where gene B has a transcript level higher than 1.25 are ecDNA(+) and otherwise they are ecDNA(-). Similarly, using Gene C, we can get perfect predictions. Thus, when Boruta has to select a gene, it will pick either Gene B or Gene C, because picking both will not improve prediction. We can therefore use Boruta to pick one gene, and then co-expression clustering to pick the other gene.

As an example, cluster #3 consists of 21 genes that were up-regulated in ecDNA(+) samples and enriched in cell-cycle related biological processes (Supplementary file 1). While these genes were expressed similarly in ecDNA(+) samples, and separately, in ecDNA(-) samples, out of the 21 genes, only 9 genes were selected in at least 10 out of 200 Boruta trials (i.e., Core genes). Of the 12 remaining genes (i.e., CorEx genes), 8 genes were not selected by the Boruta method at all, 3 genes were selected in less than 5 out of 200 Boruta trials, and 1 gene was selected in 9 out of 200 Boruta trials.

**Author response image 8. sa4fig8:** 

(4) In Fig 2a, I would like to see the variability of the precision and recall in the main text, not only the maximum values. Authors could plot mean + standard deviation for precision and recall separately, or use S2a/b.

**Response.** We have replaced Figures 2b and 2c with a combined figure (Figure 2) that gives a box-plot describing the distribution of recall values for 5 gene lists: four from the original manuscript, and another gene list created using a Generalized Linear Model (GLM).

**Author response image 9. sa4fig9:** 

(5) Since the authors analyze bulk RNA, the gene expression signatures they notice could, in principle, originate from non-tumor cells as well. I do not believe this is the case, however, the paper would be strengthened by an analysis that shows that the difference in expression patterns of the Corex genes between ecDNA(+) and ecDNA(-)-samples does come from tumor cells. One way of showing this would be by using single-cell mRNA-sequencing data, and another way of showing this would be to show that Corex gene-expression correlates with tumor purity in bulk samples.

**Response.** The reviewer is correct. Unfortunately, our analysis requires data with whole-genome sequencing (WGS) for ecDNA prediction, as well as RNA-seq for transcriptome profiling. The TCGA data-set is the only available data-set with a significant number of samples that includes both WGS and RNA-seq. They have not made tissue samples available for scRNA analysis, to our knowledge. The reviewer raises an important question regarding purity, but testing if CorEx gene expression correlates with tumor purity would require a large range of purity values, something that scientists would avoid when collecting samples.

However, the presence of non-cancer tissue (impurity) could reduce sensitivity of ecDNA detection, and therefore, change the results. To better investigate this, we started with a publication that investigated multiple tumor purity metrics and devised a composite score (CPE; Aran et al., 2015). Using their composite tumor purity, we find that ecDNA(-) samples have slightly lower purity than ecDNA(+) samples (*p*-value 0.0036; Figure 1—figure supplement 4).

This result is not surprising because one would expect lower detection of ecDNA in less pure samples. The presence of undetected ecDNA in ecDNA(-) samples would confound the results by reducing the discriminating power of genes, but would not give false results. To test this, we measured the expression directionality in CorEx genes in all samples versus samples which had a high tumor purity (CPE 0.8). The results suggest that the *p*-values of directionality in the pure samples were highly correlated with the expression data from all samples (Figure 1—figure supplement 4).

**Author response image 10. sa4fig10:** 

(6) The biological interpretation of the data remains a bit too observational. Can the authors offer an interpretation of the enriched GO terms? And are any of these genes already implicated in ecDNA(+) tumors?

**Response.** To answer the second question first, prior to our study, the focus was on genes that were amplified on ecDNA. Indeed many oncogenes known to be amplified in cancer are in fact amplified on ecDNA (Turner, Nature 2017, Kim Nature genetics 2020). This study is unique in that it identifies genes whose expression values are predictive of ecDNA(+) status. The Figure below lists 24 genes most frequently amplified on ecDNA from Kim, Nature Genetics 2020. With the exception of EGFR and CDK4, none of these 24 genes was included in the list of the 65 genes reported by us as the most frequently selected genes in the Boruta trials (lowest harmonic rank). Thus, most persistent CorEx genes do not lie on ecDNA. However, they all play important roles in biological processes relevant to cancer pathology including Immune Response, Mitotic cell Cycle, Cell division, and DSB repair. We agree with the reviewer that the results are observational (although statistically significant in populations), and some of our collaborators are actively working to experimentally validate some of these genes. The experimental work, however, is beyond the scope of this paper.

We have added the following statement to the manuscript. “Notably, of the 24 genes most frequently expressed on ecDNA,2 only EGFR and CDK4 were included in the list of 65 genes, suggesting that the most persistent CorEx genes do not themselves appear frequently on ecDNA.”

**Author response image 11. sa4fig11:** 

**Reviewer #3 (Recommendations For The Authors):**
Minor comments:(1) The authors performed gene ontology enrichment test but referred to it as gene set enrichment analysis. Usually gene set enrichment analysis does not refer to Fischer's exact test-based analysis but rather the one described in Subramanian et al 2005. The term correction should be made to avoid confusion.

**Response.** We have rephrased text in the manuscript to prevent confusion between enrichment analysis on gene sets using an one-sided Fisher’s exact test and the Gene Set Enrichment Analysis (GSEA) method that exists as a software. We have also revised the header in the methods section from “Gene set enrichment analysis” to “Gene Ontology (GO) enrichment analysis”.

(2) A couple of figures could use more detailed labels and captions. In Figure 2c, it is unclear what the numbers 100 and 54 right next to the Cliff's Delta heatmap indicate. In Figures 3a and 4a, it is not immediately clear what the barplot on top of the heatmap indicates and there is no label for the y-axis.

**Response.** These are good suggestions, and we have added descriptions to the figure captions.